# TRPM5-mediated calcium uptake regulates mucin secretion from human colon goblet cells

**Sandra Mitrovic[1], Cristina Nogueira[1], Gerard Cantero-Recasens[2], Kerstin Kiefer[2], José M Fernández-Fernández[2], Jean-François Popoff[1], Laetitia Casano[1†a], Frederic A Bard[1†b], Raul Gomez[1], Miguel A Valverde[2], Vivek Malhotra[1,3]***

[1]Department of Cell and Developmental Biology, Centre for Genomic Regulation, Barcelona, Spain; [2]Laboratory of Molecular Physiology and Channelopathies, Department of Experimental and Health Sciences, Universitat Pompeu Fabra, Barcelona, Spain; [3]Department of Cell and Developmental Biology, Catalan Institution for Research and Advanced Studies (ICREA), Barcelona, Spain

**Abstract** Mucin 5AC (MUC5AC) is secreted by goblet cells of the respiratory tract and, surprisingly, also expressed de novo in mucus secreting cancer lines. siRNA-mediated knockdown of 7343 human gene products in a human colonic cancer goblet cell line (HT29-18N2) revealed new proteins, including a $Ca^{2+}$-activated channel TRPM5, for MUC5AC secretion. TRPM5 was required for PMA and ATP-induced secretion of MUC5AC from the post-Golgi secretory granules. Stable knockdown of TRPM5 reduced a TRPM5-like current and ATP-mediated $Ca^{2+}$ signal. ATP-induced MUC5AC secretion depended strongly on $Ca^{2+}$ influx, which was markedly reduced in TRPM5 knockdown cells. The difference in ATP-induced $Ca^{2+}$ entry between control and TRPM5 knockdown cells was abrogated in the absence of extracellular $Ca^{2+}$ and by inhibition of the $Na^+/Ca^{2+}$ exchanger (NCX). Accordingly, MUC5AC secretion was reduced by inhibition of NCX. Thus TRPM5 activation by ATP couples TRPM5-mediated $Na^+$ entry to promote $Ca^{2+}$ uptake via an NCX to trigger MUC5AC secretion.

*For correspondence: vivek.malhotra@crg.eu

†Present address: [a]University of Barcelona, Faculty of Medicine, Barcelona, Spain; [b]Institute of Molecular and Cell Biology, Singapore, Singapore

## Introduction

Mucus is secreted by specialized cells that line the respiratory and digestive tract to protect against pathogens and other forms of cellular abuse. The secretion of mucus is therefore essential for the normal physiology of the wet mucosal epithelium (*Rubin, 2010*). The secretory or gel-forming mucin, Mucin 5AC (MUC5AC) is one of the major components of the mucus in the airways, and hyper- or hyposecretion of this component is a hallmark of a number of chronic obstructive pulmonary diseases (COPD) (*Rose and Voynow, 2006*). MUC5AC is also expressed at low levels in the gastrointestinal tract and, surprisingly, expressed de novo, and upregulated in colonic mucus from cancer and ulcerative colitis patients (*Bartman et al., 1999*; *Kocer et al., 2002*; *Byrd and Bresalier, 2004*; *Forgue-Lafitte et al., 2007*; *Bu et al., 2010*). MUC5AC is also expressed in response to parasitic infection, which is probably its additional physiological role (*Hasnain et al., 2011*).

The gel-forming mucins are giant filamentous glycoproteins that are synthesized in the Endoplasmic Reticulum (ER) and exported to the Golgi complex where they undergo extensive modification in their oligosaccharide chains. The apparent molecular weight of the gel-forming MUC5AC increases from 500 kD of monomeric unglycosylated ER form (*van Klinken et al., 1998*) to 2.2 MD (*Thornton et al., 1996*) by glycosylation and oligomerization during its transit through the Golgi apparatus to a secreted form that reaches up to 40 MD in apparent molecular weight (*Sheehan et al., 2000*). The heavily

**eLife digest** Goblet cells are specialized cells that produce proteins called mucins, which combine with water, salt and other proteins to form mucus, the slippery fluid that protects the respiratory and digestive tracts from bacteria, viruses and other pathogens. However, a defect in the production of one particular type of mucin—Mucin 5AC—can result in diseases such as cystic fibrosis, chronic obstructive pulmonary disease and Crohn's disease, so there is a clear need to understand the production of mucus in detail.

Before they are secreted, the mucins are packaged inside granules in the goblet cells. When a certain extracellular signal arrives at a goblet cell, these granules move through the cell, fuse with the cell membrane and release the mucins, which then expand their volume by a factor of up to a 1000. Calcium ions ($Ca^{2+}$) have a critical role in the signal that leads to the secretion of mucins, but many details about the signalling and secretion processes are poorly understood.

Now, Mitrovic et al. have used genetic methods to study 7343 gene products in goblet cells derived from a human colon. They identified 16 new proteins that are involved in the secretion of Mucin 5AC, including a channel protein called TRPM5. This protein is activated when the concentration of $Ca^{2+}$ inside the cell increases, and its activation allows sodium ($Na^+$) ions to enter the cells. These intracellular $Na^+$ ions are then exchanged for $Ca^{2+}$ ions from outside the cell, and these $Ca^{2+}$ ions then couple to the molecular machinery that is responsible for the secretion of the mucins.

By using electrophysiological and $Ca^{2+}$ imaging approaches, Mitrovic et al. were able to visualize and measure TRPM5-mediated $Na^+$ currents and the subsequent $Ca^{2+}$ uptake by the cells, and confirmed that extracellular $Ca^{2+}$ ions were responsible for stimulating the secretion of mucins. The next step is to determine how the other 15 genes are involved in mucin secretion and, in the longer term, explore how these insights might be translated into treatments for cystic fibrosis and other conditions associated with defective mucus secretion.

glycosylated mucins are sorted, condensed and packed into mucin-secreting granules (MSG). The MSG fuse with the plasma membrane, in a signal-dependent manner, and the condensed mucins expand their volume up to 1000-fold upon secretion (*Verdugo, 1993*).

The signaling events that lead to mucin secretion in the airways involve mainly, but not exclusively, P2Y purinergic and muscarinic receptor activation by ATP and acetylcholine, respectively. The subsequent generation of diacylglycerol (DAG) and inositol 1, 4, 5-triphosphate (IP3) activate protein kinase-C (PKC) and cause the release of $Ca^{2+}$ from the ER to promote mucus secretion (*Bou-Hanna et al., 1994*; *Abdullah et al., 1996*, *1997*; *Bertrand et al., 2004*; *Ehre et al., 2007*).

The progress to date on the components involved in the trafficking of mucins has recently been thoroughly reviewed (*Davis and Dickey, 2008*). Basically, mucins are packed (somehow) into MSG at the *trans*-Golgi network (TGN). MSGs undergo fusion to produce mature condensed granules that are stored in the cytoplasm. The cortical actin acts as a barrier that is reorganized in a $Ca^{2+}$-dependent reaction through the input of PKCε-dependent phosphorylation of MARCKS (*Wollman and Meyer, 2012*). The passage of mature MSGs through the actin network also requires Myo II and V. The proteins involved in the docking, priming and fusion of the MSGs are reported to include: Rab3d, Rab27, Hsc70, cysteine string protein, Synaptotagmin 2, Munc13-2, Munc13-4, Munc18b, Syntaxin 2, 3, 11, and VAMP8. However, it is not known how many of these proteins are directly involved in mucin secretion and for some, such as the MARCKS protein, the mechanism is controversial (*Stumpo et al., 1995*; *Arbuzova et al., 2002*). The exact myosin involved in the trafficking of MSGs across the actin barrier remains unclear (*Rose et al., 2003*; *Neco et al., 2004*; *Jerdeva et al., 2005*). More importantly, the mechanism of $Ca^{2+}$-dependent signaling and the components involved in this signaling cascade are not fully characterized.

To date, transport studies have been based on truncated GFP-mucin variants (*Perez-Vilar et al., 2005*) and time-consuming techniques such as combinations of density gradient centrifugation and agarose gel electrophoresis (*Sheehan et al., 2004*). It has therefore been difficult to identify new components involved in mucin secretion and to decipher their mechanism of action.

As stated above, human cancer cells and cells from patients with ulcerative colitis express and secrete MUC5AC. These cells and cell lines therefore provide a convenient means to address the mechanism MUC5AC secretion. We have established a quantitative assay to measure the secretion of MUC5AC from a human goblet cell line. The procedure was used to screen 7343 human gene products and we describe here the identification and involvement of transient receptor potential melastatin 5 (TRPM5) channel in MUC5AC secretion.

## Results

### An assay for mucin secretion

The human colonic adenocarcinoma cells HT29-18N2 (N2) differentiate to goblet cells upon starvation in protein-free medium (*Phillips et al., 1995*), which increases the production of MUC5AC. Immunofluorescence analysis of accumulated MUC5AC in secretory granules (*Figure 1A*) shows the differences between starved and nonstarved cells. The increase in protein production of MUC5AC after starvation was confirmed by dot-blotting cell lysates of nonstarved and starved N2 cells (*Figure 1B*). Quantification of the dot blot revealed a 45-fold increase of MUC5AC protein levels in starved N2 cells compared to nonstarved N2 cells. Our findings with the dot-blot procedure confirm the lack of MUC5AC production in Hela cells (*Figure 1B,C*). MUC5AC mRNA analysis by quantitative real-time PCR also confirmed increased MUC5AC mRNA levels in starved cells (*Figure 1D*). The fusion of MUC5AC-containing granules with the plasma membrane requires an external signal, which results in the production of DAG and the release of $Ca^{2+}$ from internal stores. To induce mucin secretion from the starved N2 cells, we used the DAG mimic, phorbol-12-myristate-13-acetate (PMA). Starved goblet cells were treated for 2 hr with 2 μM PMA to induce MUC5AC secretion (*Figure 1E*). The extracellular MUC5AC expands and coats the cell surface (*Figure 1E*). We took advantage of the stickiness of the mucin film to quantitate secreted MUC5AC. After 2 hr incubation with PMA, the cells were fixed with paraformaldehyde followed by incubation with an anti-MUC5AC antibody and a secondary fluorescent-labeled antibody to visualize secreted mucin (*Figure 1E*). To detect the intracellular pool of MUC5AC after PMA-induced release, the cells were washed extensively to remove secreted MUC5AC and then fixed with paraformaldehyde, permeabilized and processed for immunofluorescence microscopy with an anti-MUC5AC antibody as described above (*Figure 1E*).

To quantitate MUC5AC secretion, starved goblet cells were treated for 2 hr with 2 μM PMA, followed by fixation and incubation with an anti-MUC5AC antibody. The secreted MUC5AC was monitored by chemiluminescence using secondary antibodies conjugated to HRP (*Figure 2A,B*). The time course for PMA induced MUC5AC secretion shows a significant increase at 15 min and maximal MUC5AC secretion is observed at 2 hr post incubation with 2 μM PMA (*Figure 2—figure supplement 1*).

Secretion of mucins requires a dynamic actin cytoskeleton and $Ca^{2+}$ (*Abdullah et al., 1997*; *Ehre et al., 2005*; *Wollman and Meyer, 2012*). We tested the effect of perturbing actin cytoskeleton and $Ca^{2+}$ levels on the PMA-dependent secretion of MUC5AC from starved N2 cells. Starved N2 cells were treated with the drugs that affect actin filaments: Latrunculin A and Jasplakinolide. The cells were also treated with the membrane-permeant $Ca^{2+}$ chelator BAPTA-AM. The extracellular levels of MUC5AC were measured with the chemiluminescence-based assay. Depolymerization of actin filaments by Latrunculin A had no effect on PMA-stimulated MUC5AC secretion, while BAPTA-AM and the actin-stabilizing agent Jasplakinolide severely affected MUC5AC secretion (*Figure 2C*). The inhibitory effect of hyperstabilized actin filaments (by Jasplakinolide treatment) on MUC5AC secretion reveals that actin filaments likely act as a barrier to prevent premature fusion of MUC5AC-containing granules with the cell surface. Inhibition of MUC5AC secretion by BAPTA-AM treatment confirms the known requirement of $Ca^{2+}$ in the events leading to mucin secretion.

### PMA induces the release of post-Golgi pool of MUC5AC

Befreldin A (BFA) is known to inhibit cargo export from the ER and causes Golgi membranes to fuse with the ER (*Lippincott-Schwartz et al., 1989*). To test whether BFA affected the formation of secretory granules, starved N2 cells were incubated with or without 2 μg/ml BFA. After 45 min cells were fixed and examined by immunofluorescence analysis with an anti-MUC5AC antibody and an antibody to the Golgi membrane specific GRASP65 protein (*Figure 2E*). The dispersal of GRASP65 with BFA treatment shows that our experimental conditions are effective in disrupting the Golgi apparatus. However, MUC5AC staining was unperturbed by BFA treatment (*Figure 2E*). We then tested the effect of BFA

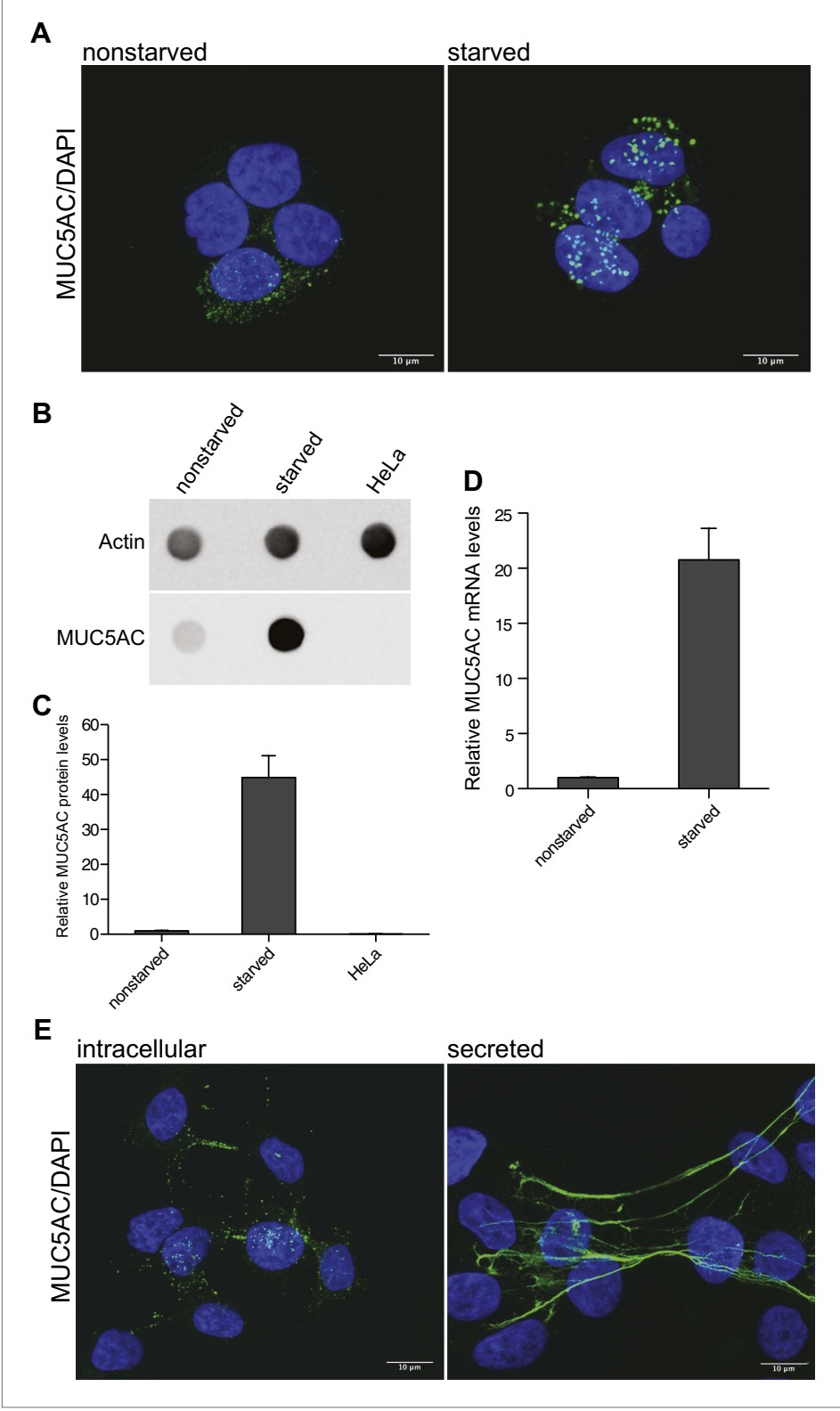

**Figure 1**. Mucin synthesis and secretion from goblet cells. (**A**) Nonstarved and starved N2 cells were fixed and analyzed by immunofluorescence microscopy with an anti-MUC5AC antibody (green). The nuclear DNA was stained with DAPI (blue) to localize the position of the nucleus. (**B**) Dot blot of total lysates of nonstarved, starved N2 and HeLa cells were probed with anti-MUC5AC and anti-actin antibody. (**C**) The dot blots in (**B**) were quantified

*Figure 1. Continued on next page*

*Figure 1. Continued*

and normalized to actin levels. The y-axis represents relative values with respect to the values of nonstarved N2 cells. Average values ± SEM are plotted as bar graphs (N = 3). (**D**) Nonstarved and 5 days starved N2 cells were lysed and total RNA was extracted for quantitative real-time PCR analysis. The values for MUC5AC mRNA levels were normalized to the values of the housekeeping gene HPRT1. The y-axis represents relative values with respect to nonstarved N2 cells. Average values ± SEM are plotted as bar graphs (N = 4). (**E**) Starved N2 cells were treated for 2 hr with 2 µM PMA. To detect the remaining intracellular mucin after PMA release, the secreted mucin was removed by DTT and trypsin treatment of the goblet cells prior to fixation (experimental procedures). After fixation, cells were permeabilized and examined by immunofluorescence microscopy with DAPI and an anti-MUC5AC antibody. Secreted MUC5AC was detected by fixing the secreted mucus directly on the cells after PMA treatment, followed by immunofluorescence microscopy using an anti-MUC5AC specific antibody.

on the constitutive secretion of newly synthesized proteins. Starved N2 cells were labeled with [35]S-methionine and then chased in cold methionine-containing medium in the presence of BFA. Analysis of the medium revealed that BFA severely inhibited the secretion of newly synthesized proteins from the starved N2 cells (*Figure 3—figure supplement 1*). To test whether BFA affected the regulated secretion of the secretory granules, starved N2 cells were pretreated with 2 µg/ml BFA for 15 min and then treated with 2 µM PMA for 2 hr in the presence of BFA. MUC5AC was then measured in the extracellular medium by chemiluminescence (*Figure 2D*). The results reveal that BFA treatment does not affect PMA-dependent MUC5AC secretion under the experimental conditions. Therefore, in our assay, we only measure the secretion of MUC5AC contained in the post-Golgi secretory carriers. This measurement is independent of MUC5AC synthesis, export from the ER to the late Golgi, and its sorting and packing into the secretory granules.

## Identification of proteins involved in mucin secretion (PIMS)

N2 cells were starved for 6 days and transfected with siRNA oligos against each of the selected 7343 genes. A pool of four different siRNAs targeting the same component was used and every component was analyzed in triplicate. 3 days after transfection, the cells were treated with 2 µM PMA for 2 hr and analyzed by chemiluminescence-based detection of secreted MUC5AC (*Figures 2A and 3A*). For the data analysis we assumed that the majority of the siRNAs will not affect the secretion of MUC5AC. Data points were normalized by the B-score and the triplicates were ranked according to the Ranking Product method (*Breitling et al., 2004*; *Supplementary file 1*). The hits were plotted as median of the B-score and positive hits were selected above and below a B-score of ±1.5. siRNAs that scored above 1.5 B-score were considered as hypersecretory phenotype and those below 1.5 B-score were considered as inhibitors of MUC5AC secretion (hyposecretory phenotype) (*Figure 3B*). From this analysis we selected 413 components that upon knockdown resulted in hyposecretion and 534 that exhibited a hypersecretion of MUC5AC (*Figure 3C*). The hits were analyzed by Ingenuity Pathway Analysis and categorized according to their intracellular localization and type. For further analysis we removed 678 proteins from this pool that included secreted proteins, nuclear proteins, proteins affecting protein modification, and those involved in basic metabolism. This narrowed the hits to 114 with hyposecretion and 155 with hypersecretion phenotype (*Supplementary file 1*).

This collection of 269 hits was rescreened with another siRNA library composed of a pool of four different siRNAs targeting the same protein. The same procedure as described above was used to monitor the effect of siRNA on MUC5AC secretion. The secreted MUC5AC levels were normalized with the Z-score. For the hit analysis we assumed mainly positive hits affecting MUC5AC secretion. Therefore the cutoff was set according to mock transfected cells ±2 SD. With that setup, we identified 29 components exhibiting a hyposecretory phenotype and 5 with a hypersecretory phenotype (*Figure 3C* and *Table 1*).

It is important to test whether any of the proteins identified in our screening assay have a role in constitutive secretion of cargoes that do not enter the secretory granules. This could reveal the convergent function of PIMS in conventional and regulated protein secretion. N2 cells were starved for 6 days, transfected with siRNAs for the individual PIMS, and 3 days later were washed in methionine free medium for 20 min. The cells were then incubated with [35]S-methionine containing medium for 15 min and subsequently cultured in methionine containing medium. After 3 hr, the medium was collected and the cells were lysed and measured for total [35]S-methionine incorporation. As a control,

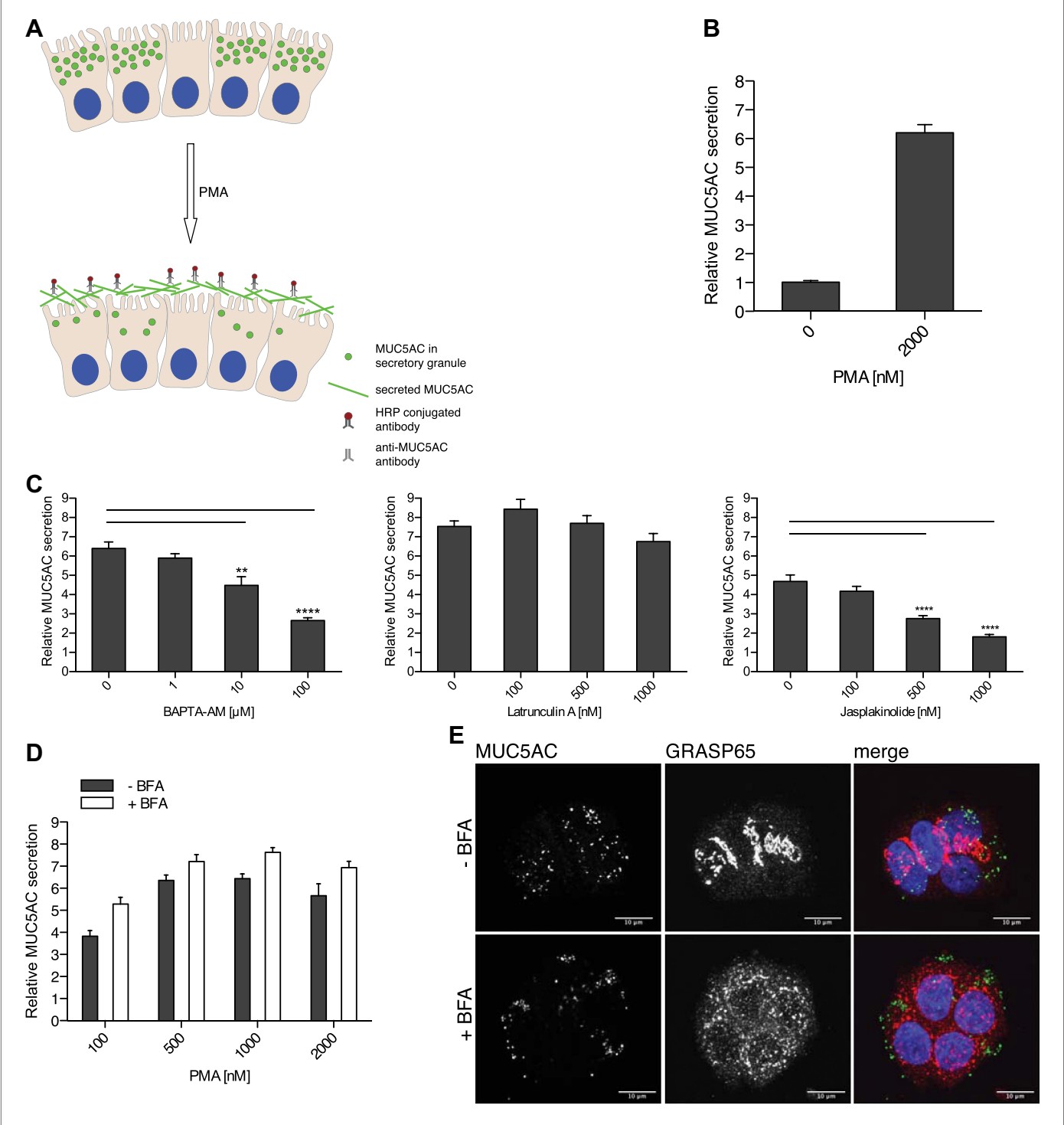

**Figure 2**. Mucin secretion assay. (**A**) Illustration of the mucin secretion assay. Starved N2 cells are treated with PMA. Secreted MUC5AC is fixed on the cell surface and labeled with anti-MUC5AC antibodies followed by quantitative detection using HRP-conjugated secondary antibody. (**B**) Starved N2 cells were treated for 2 hr ± 2 µM PMA, fixed with formaldehyde and the amount of secreted MUC5AC bound to the cell surface was detected with anti-MUC5AC antibody and measured by chemiluminescence. The values were normalized to values obtained for—PMA treatment. Average values ± SEM are plotted as bar graphs (N = 10). (**C**) Starved N2 cells were pretreated for 30 min with BAPTA-AM, Latrunculin A or Jasplakinolide and incubated at 37°C. After 30 min, 2 µM PMA was added containing the respective drugs (BAPTA-AM, Latrunculin A or Jasplakinolide) and incubated for 2 hr at 37°C. Cells were fixed and secreted MUC5AC was detected by chemiluminescence. The values were normalized to values obtained for −PMA treatment.
*Figure 2. Continued on next page*

*Figure 2. Continued*

Average values ± SEM are plotted as bar graphs (N = 10). Compared datasets were considered as statistically significant when p<0.01 (**) and p<0.0001 (****).
(**D**) Starved N2 cells were preincubated for 15 min with 2 µg/ml (+BFA) and incubated at 37°C. After 30 min, increasing concentrations of PMA were added in the presence or absence of 2 µg/ml BFA (+/− BFA) and incubated for 2 hr at 37°C. Cells were fixed and secreted MUC5AC was detected by chemiluminescence. The values were normalized to values obtained for −PMA treatment. Average values ± SEM are plotted as bar graphs (N = 5).
(**E**) Starved N2 cells were incubated for 45 min with or without 2 µg/ml BFA (+/− BFA) at 37°C. After 45 min cells were fixed, permeabilized and examined by immunofluorescence microscopy with an anti-MUC5AC antibody (green), an anti-GRASP65 antibody (red) and DAPI (blue).

The following figure supplements are available for figure 2:

**Figure supplement 1**. Time course for MUC5AC secretion.

BFA was added (at 2 µg/ml) during the whole pulse-chase procedure to inhibit general protein secretion. The secreted medium was normalized to the incorporation of $^{35}$S-methionine, corresponding volumes precipitated by TCA, and analysis done by SDS-PAGE. We did not detect any obvious changes in the secreted polypeptide pool in starved N2 cells depleted of the individual PIMS compared with control cells (*Figure 3—figure supplement 1*). As expected, BFA treatment inhibited general protein secretion (*Figure 3—figure supplement 1*).

## Expression of proteins involved in mucin secretion in N2 cells

We tested whether the proteins selected for MUC5AC secretion are expressed in starved and unstarved cells, as their expression could depend on the differentiation state of the cells. Total RNA was extracted from starved and unstarved N2 cells and the expression of 34 components selected for MUC5AC secretion (*Table 1*) was determined by reverse transcription of the mRNA and PCR with specific primers for each component. The PCR products were separated by agarose gel electrophoresis, quantified and normalized to the housekeeping gene GAPDH (*Figure 4A,B*). Our findings reveal that 16 of the 34 hits can be detected in N2 cells (*Table 1* and *Figure 4A*). Of these, the mRNA levels of eight components remain unchanged upon starvation compared with unstarved cells (*Figure 4A,B*). Interestingly, in differentiated N2 cells the levels of GRIK4, KCNIP3, SIPA1, MAPK15, ATF6 and CCR9 were upregulated, while the mRNA levels of SUR1 and SILV were downregulated (*Figure 4A,B*). As expected, the MUC5AC mRNA was upregulated upon starvation whereas the mRNA levels for the housekeeping gene GAPDH remained unchanged (*Figure 4A,B*). Based on the data presented thus far we suggest that these 16 proteins are required for MUC5AC secretion and have labelled them as PIMS (Protein Involved in Mucin Secretion) for the sake of simplicity (*Table 2*).

## TRPM5 is required for PMA-induced MUC5AC secretion

TRPM5, a hit in our assay, is required for the regulated secretion of insulin (*Brixel et al., 2010*; *Colsoul et al., 2010*) and this prompted us to further test its involvement in the regulated secretion of MUC5AC. TRPM5 is a $Ca^{2+}$-activated monovalent cation-selective channel that responds to warm temperature and participates in the taste-receptor signaling pathway (*Perez et al., 2002*; *Hofmann et al., 2003*; *Zhang et al., 2003*; *Talavera et al., 2005*). We generated a shRNA-dependent stable N2 goblet cell line depleted of TRPM5. This procedure resulted in greater than 80% reduction in the TRPM5 mRNA levels (*Figure 5A*) and a 50% reduction in the quantity of PMA dependent MUC5AC secreted by starved cells (*Figure 5B*). The defect in secretion could reflect a reduction in the total intracellular protein pool of MUC5AC. Therefore, we probed the total cell lysate from TRPM5 knockdown and control cells with an anti-MUC5AC antibody by dot blot. The dot blot was quantified and revealed no difference in the total MUC5AC contents in the TRPM5 knockdown cells compared with control cells (*Figure 5C*). We also tested whether the reduction in secretion is due to a defect in granule formation. Intracellular granules were visualized by immunofluorescence microscopy with an anti-MUC5AC antibody in starved TRPM5-depleted cells. GFP-positive cells indicate the expression of TRPM5 shRNA and show that secretory granules were formed in TRPM5-depleted cells upon starvation (*Figure 5D*). As shown in *Figure 2D*, secretion of MUC5AC is BFA-independent and therefore from a post-Golgi pool.

## Location and activity of TRPM5 in N2 goblet cells

HA-tagged (N-terminal) TRPM5 was expressed in N2 cells and localized by confocal fluorescence microscopy in cells grown as polarized monolayers. HA-TRPM5 was localized to the apical surface of

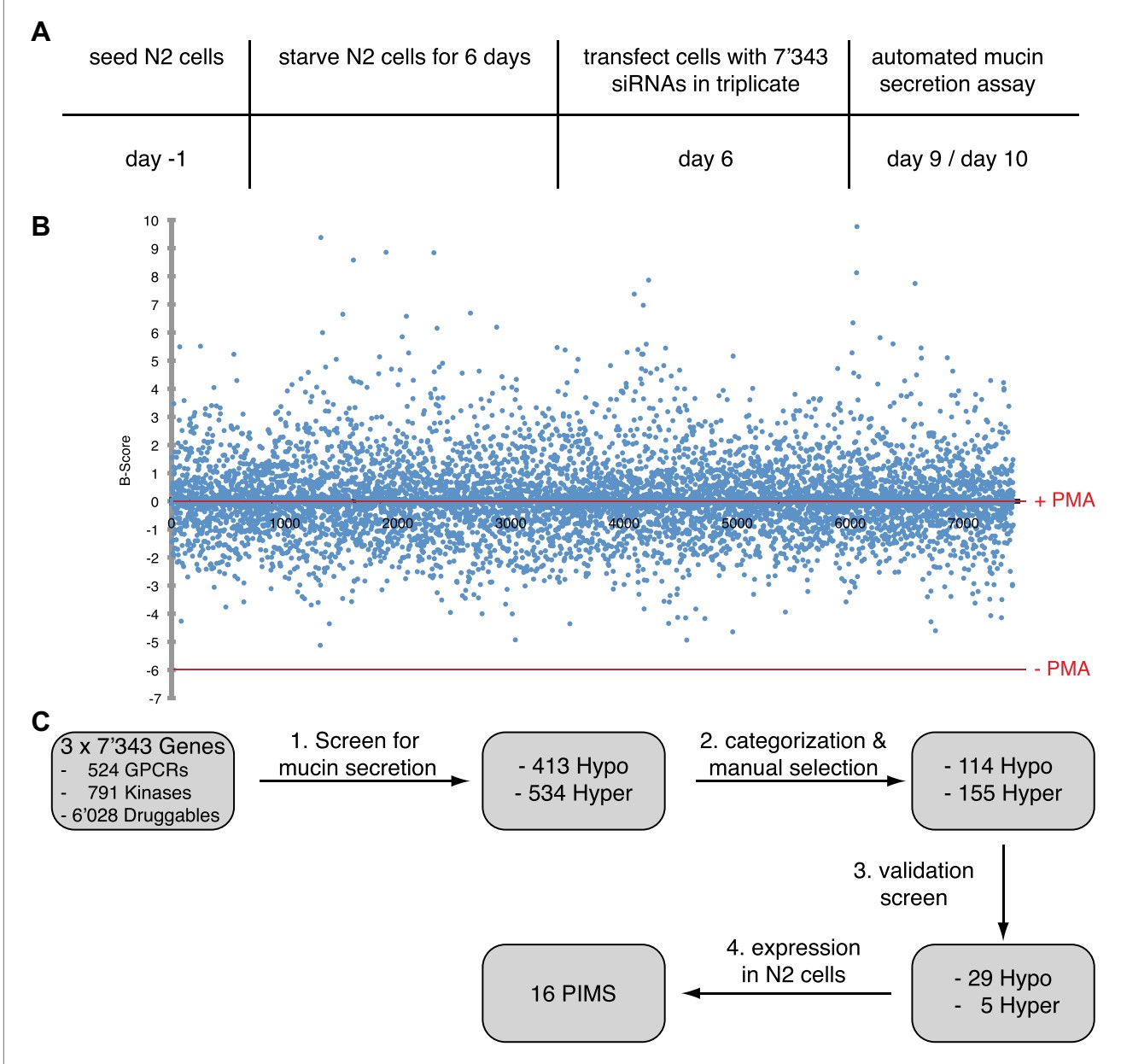

**Figure 3**. Identification of proteins required for MUC5AC secretion. (**A**) Screening procedure. N2 cells were serum starved for 6 days. The cells were then seeded into the wells of 96-well plates and transfected in triplicates on three sets of plates with siRNAs directed against 7343 components. 3 days after transfection, the cells were treated for 2 hr with 2 μM PMA and analyzed by an automated chemiluminescence assay for the detection of secreted MUC5AC. (**B**) The B-score of each gene product tested was calculated using the triplicate measurements of the chemiluminescence values. All siRNAs that altered secretion with a B-score less than −1.5 (hyposecretory) and higher than 1.5 (hypersecretory) were selected as positive hits for further analysis. −PMA (lower red line) and +PMA (upper red line) indicate mock treated controls without PMA (−PMA) and with 2 μM PMA (+PMA). (**C**) Flowchart depicting screen for PIMS and validation process. From the initial collection of 7343 siRNAs, 413 were classified as hyposecretory and 534 as hypersecretory hits (***Supplementary file 1***). From this pool we further selected 114 hyposecretory and 155 hypersecretory components (***Supplementary file 1***). A second validation screen with siRNAs distinct from the primary screen confirmed 29 proteins that gave a MUC5AC hyposecretory phenotype and 5 with a hypersecretory secretory defect (***Table 1***).

The following figure supplements are available for figure 3:

**Figure supplement 1**. PIMS are not required for constitutive secretion of newly synthesized proteins.

**Table 1.** Thirty-four proteins involved in MUC5AC secretion

| Gene ID | Symbol | MUC5AC secretion | Type | Known localization | Expression in N2 cells |
|---|---|---|---|---|---|
| 6833 | SUR1 | Hyposecretory | Transporter | PM | Detected |
| 2900 | GRIK4 | Hyposecretory | Ion channel | PM | Detected |
| 29,850 | TRPM5 | Hyposecretory | Ion channel | PM | Detected |
| 6326 | SCN2A | Hyposecretory | Ion channel | PM | Not detected |
| 6328 | SCN3A | Hyposecretory | Ion channel | PM | Not detected |
| 6915 | TBXA2R | Hyposecretory | GPCR | PM | Not detected |
| 6753 | SSTR3 | Hyposecretory | GPCR | PM | Not detected |
| 122042 | RXFP2 | Hyposecretory | GPCR | PM | Not detected |
| 1394 | CRHR1 | Hyposecretory | GPCR | PM | Not detected |
| 10,803 | CCR9 | Hyposecretory | GPCR | PM | Detected |
| 524 | CX3CR1 | Hyposecretory | GPCR | PM | Not detected |
| 8484 | GALR3 | Hyposecretory | GPCR | PM | Not detected |
| 83,550 | GPR101 | Hyposecretory | GPCR | PM | Not detected |
| 118442 | GPR62 | Hyposecretory | GPCR | PM | Detected |
| 4914 | NTRK1 | Hyposecretory | TM receptor | PM | Not detected |
| 9437 | NCR1 | Hyposecretory | TM receptor | PM | Not detected |
| 55,824 | PAG1 | Hyposecretory | Other | PM | Detected |
| 259215 | LY6G6F | Hyposecretory | TM receptor | PM | Not detected |
| 56,140 | PCDHA8 | Hyposecretory | Cadherin | PM | Not detected |
| 225689 | MAPK15 | Hyposecretory | Kinase | Cytosol | Detected |
| 23,542 | MAPK8IP2 | Hyposecretory | Other | Cytosol | Not detected |
| 10,454 | TAB1 | Hyposecretory | Scaffold | Cytosol | Detected |
| 6490 | SILV | Hyposecretory | Melanosome biogenesis | Melanosome | Detected |
| 6494 | SIPA1 | Hyposecretory | GTPase activator | Cytosol, nucleus | Detected |
| 30,818 | KCNIP3 | Hyposecretory | Ca$^{2+}$ binding | Palmitoylated: membrane, cytosol | Detected |
| 22,926 | ATF6 | Hyposecretory | Transcription | ER | Detected |
| 26,499 | PLEK2 | Hyposecretory | Actin binding | Cytoskeleton | Detected |
| 4790 | NFKB1 | Hyposecretory | Transcription | Nucleus, cytosol | Detected |
| 6720 | SREBF1 | Hyposecretory | Transcription | ER | Detected |
| 1238 | CCBP2 | Hypersecretory | GPCR | PM | Detected |
| 84,033 | OBSCN | Hypersecretory | Kinase | Cytosol | NT |
| 9373 | PLAA | Hypersecretory | Other | Cytosol | NT |
| 4843 | NOS2 | Hypersecretory | Other | Cytosol | NT |
| 4893 | NRAS | Hypersecretory | G-protein | Cytosol | NT |

ER: endoplasmic reticulum; GPCR: G protein coupled receptor; PM: plasma membrane; TM receptor: transmembrane receptor; NT: not tested.

the polarized non-starved and starved N2 cells while apical MUC5AC staining was only evident in starved N2 cells (*Figure 6A*). TRPM5 channel (green, *Figure 6A*) appears proximal to the nucleus in non-starved cells because these cells are shorter in height compared with the starved-differentiated cells. Next, we checked the presence of functional TRPM5 channels at the plasma membrane of starved N2 cells. For this purpose, we carried out whole-cell patch-clamp experiments. Dialyzing N2 cells with Ca$^{2+}$-free intracellular pipette solutions resulted in negligibly low cationic current

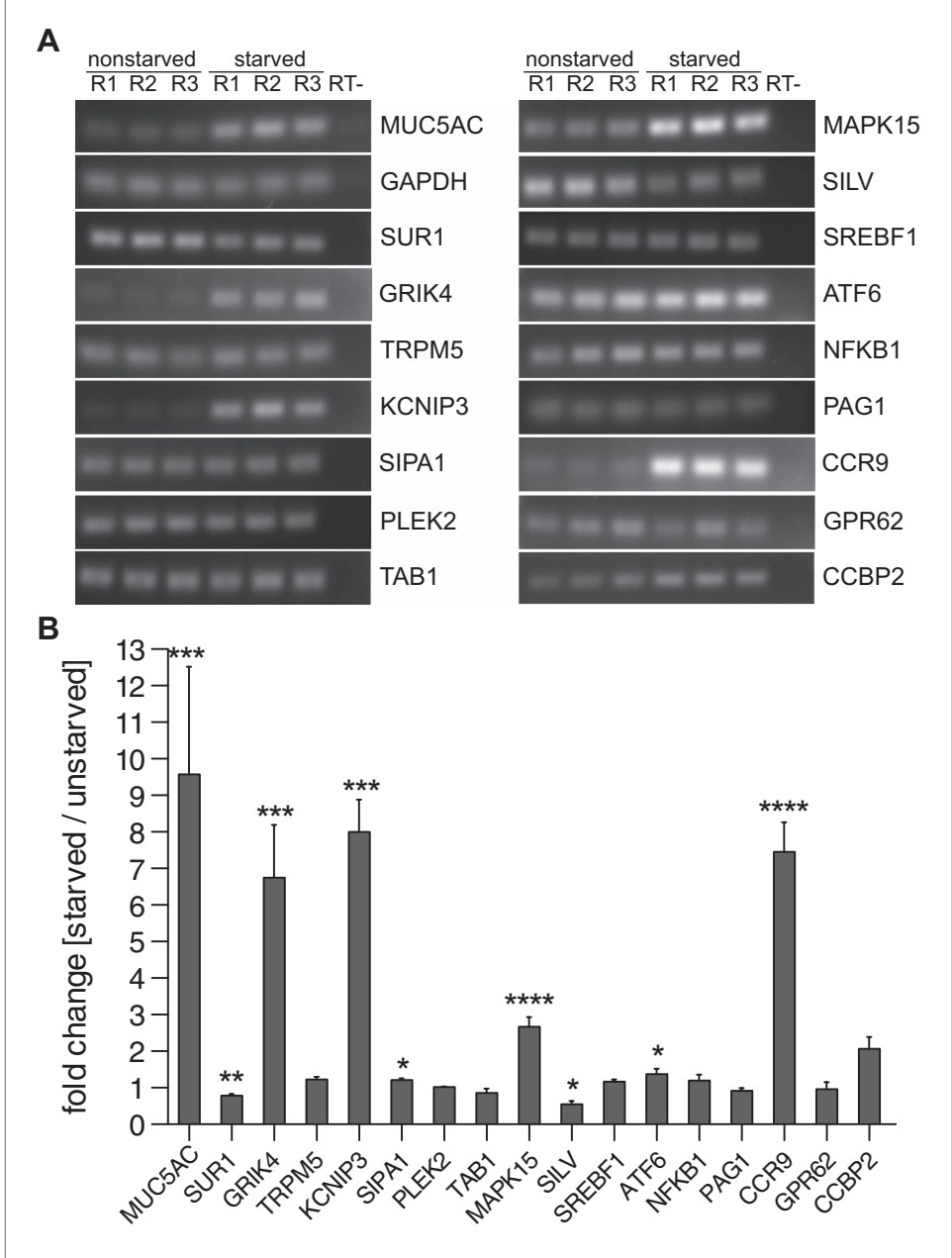

**Figure 4**. Expression profile of PIMS. (**A**) Total RNA was extracted from nonstarved- and 5-day starved N2 cells. 1 µg of total RNA was used to generate cDNA by reverse transcription. PCR was performed on cDNA with primers specific for the indicated genes and PCR products were analyzed by agarose gel electrophoresis. (**B**) Results in (**A**) were quantified and values were normalized to the housekeeping gene GAPDH. The y-axis represents relative values of starved compared to nonstarved N2 cells. Average values ± SEM are plotted as bar graphs (N = 3). Compared datasets were considered as statistically significant when $p < 0.05$ (*), $p < 0.01$ (**), $p < 0.001$ (***) and $p < 0.0001$ (****). Abbreviations: R1: replicate 1; R2: replicate 2; R3: replicate 3; RT-: reverse transcription without reverse transcriptase.

recorded; the presence of 1 µM $Ca^{2+}$ in the pipette solution generated an outwardly rectifying TRPM5-like current (***Figure 6B***). Replacement of extracellular $Na^+$ by the non-permeant cation N-methyl-D-glucamine reduced the inward current without affecting the outward current (***Figure 6B***). shRNA-mediated knockdown of TRPM5 abolished $Ca^{2+}$-dependent cationic currents recorded in starved N2 cells (***Figure 6C,D***). Together, these results are consistent with the participation of the TRPM5 channel in the generation of the plasma membrane, nonselective, $Ca^{2+}$-dependent cationic current in N2 cells.

**Table 2.** Identification of proteins involved in MUC5AC secretion (PIMS)

| Gene | PIMS | MUC5AC secretion | Type | Known localization | Expression in starved N2 cells | Disease |
|---|---|---|---|---|---|---|
| SUR1 | 1 | Hyposecretory | Transporter | PM | Unchanged | Diabetes mellitus, insulin secretion (*Aittoniemi et al., 2009*) |
| GRIK4 | 2 | Hyposecretory | Ion channel | PM | Upregulated | |
| TRPM5 | 3 | Hyposecretory | Ion channel | PM | Unchanged | Insulin secretion (*Colsoul et al., 2010*) |
| KCNIP3 | 4 | Hyposecretory | Ca$^{2+}$ binding | Palmitoylated membrane, cytosol | Upregulated | |
| SIPA1 | 5 | Hyposecretory | GTPase activator | Cytosol, nucleus | Unchanged | |
| PLEK2 | 6 | Hyposecretory | Actin binding | Cytoskeleton | Unchanged | |
| TAB1 | 7 | Hyposecretory | Scaffold | Cytosol | Unchanged | |
| MAPK15 | 8 | Hyposecretory | Kinase | Cytosol | Upregulated | |
| SILV | 9 | Hyposecretory | Melanosome biogenesis | Melanosome | Unchanged | |
| SREBF1 | 10 | Hyposecretory | Transcription regulator | ER | Unchanged | |
| ATF6 | 11 | Hyposecretory | Transcription regulator | ER | Unchanged | |
| NFKB1 | 12 | Hyposecretory | Transcription regulator | Cytosol, nucleus | Unchanged | MUC5AC biosynthesis (*Fujisawa et al., 2009*); asthma (*Hart et al., 1998*); COPD (*di Stefano et al., 2002*) |
| PAG1 | 13 | Hyposecretory | Other | PM | Unchanged | |
| CCR9 | 14 | Hyposecretory | GPCR | PM | Upregulated | Inflammatory bowel disease (*Nishimura et al., 2009*) |
| GPR62 | 15 | Hyposecretory | GPCR | PM | Unchanged | |
| CCBP2 | 16 | Hypersecretory | GPCR | PM | Unchanged | |

ER: endoplasmic reticulum; GPCR: G protein-coupled receptor; PM: plasma membrane; TM receptor: transmembrane receptor.

## TRPM5 is required for ATP-induced MUC5AC secretion

Is TRPM5 required for goblet cell response to physiological secretagogues such as ATP? The induction of mucin secretion by ATP is linked to phospholipase C activation, IP$_3$ and DAG generation, and Ca$^{2+}$ release from the ER (*Abdullah et al., 2003*; *Davis and Dickey, 2008*). To test the involvement of TRPM5 in ATP-induced MUC5AC secretion, wild-type and stable TRPM5-depleted N2 cells were differentiated and treated with ATP for 30 min. Knockdown of TRPM5 significantly reduced ATP-induced secretion of MUC5AC (*Figure 7A*). These findings strongly support the significance of our procedure for the identification of proteins required for the regulated secretion of MUC5AC and confirm the role of TRPM5 in mucin secretion under physiological (extracellular ATP) conditions.

Ca$^{2+}$ entry in response to different secretagogues has been reported for several types of mucin-secreting cells, although the mechanism of Ca$^{2+}$ influx is poorly understood (*Bou-Hanna et al., 1994*; *Bertrand et al., 2004*; *Lu et al., 2011*). Since Ca$^{2+}$ is essential for the regulated secretion of MUC5AC and TRPM5 channel activity is known to affect cell membrane potential, we reasoned that the role of

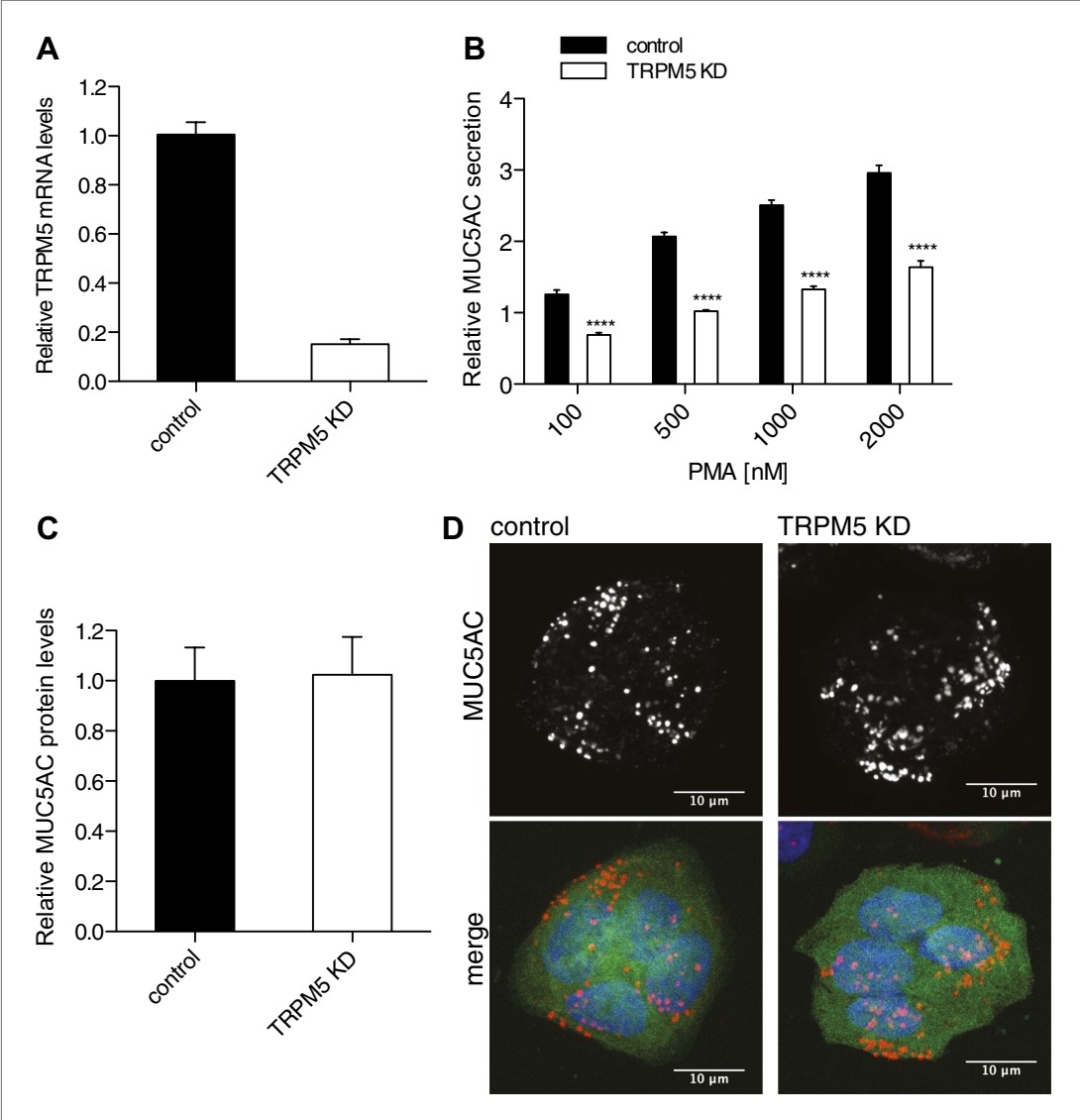

**Figure 5**. TRPM5 in mucin homeostasis. (**A**) Total RNA was extracted from control and TRPM5 stable knockdown (TRPM5 KD) cells and analyzed for knockdown efficiency of TRPM5 on mRNA level by quantitative real-time PCR using primers specific for TRPM5. TRPM5 values were normalized to values of the housekeeping gene GAPDH. The knockdown of TRPM5 is represented as relative value compared to control cells. Results are means ± SEM. (N = 5). (**B**) Control and TRPM5 stable knockdown cells were starved for 6 days and seeded on 96-well plates. At day 9, cells were treated with increasing concentrations of PMA for 30 min and analyzed by chemiluminescence using anti-MUC5AC antibody. After the mucin secretion assay these cells were stained with DAPI and imaged by fluorescence microscopy. Nuclei were counted using ImageJ and the chemiluminescence value for MUC5AC of each well was normalized to the number of nuclei per well. The y-axis represents relative values with respect to the values of control cells not treated with PMA. Average values ± SEM are plotted as bar graphs (N = 5). Datasets were considered as statistically significant when p<0.0001 (****). (**C**) N2 cells were starved for 6 days and seeded for dot blot analysis. At day 9 cells were lysed and analyzed by dot blot with an anti-MUC5AC and anti-actin antibody. The intensity of the spots was quantified using ImageJ. Intensities of MUC5AC spots were normalized to the intensity of actin spots. Results are means ± SEM. (N = 6). (**D**) Control and TRPM5 stable knockdown cells were differentiated by starvation. After starvation cells were fixed, permeabilized and analyzed by immunofluorescence microscopy with an anti-MUC5AC antibody (red) and DAPI (blue). Cells shown in green represent expression of GFP, showing that these cells express shRNA specific for TRPM5 and are depleted of TRPM5.

TRPM5 in MUC5AC secretion is to modulate secretagogue-induced $Ca^{2+}$ influx. We tested whether extracellular $Ca^{2+}$ was required for ATP-dependent MUC5AC secretion. N2 cells were differentiated by starvation and washed; MUC5AC secretion was then induced in either $Ca^{2+}$-containing or $Ca^{2+}$-free medium for 30 min in the presence of 100 μM ATP. N2 cells cultured without $Ca^{2+}$ in the extracellular medium did not secrete MUC5AC upon exposure to ATP. However, there was a sixfold increase in MUC5AC

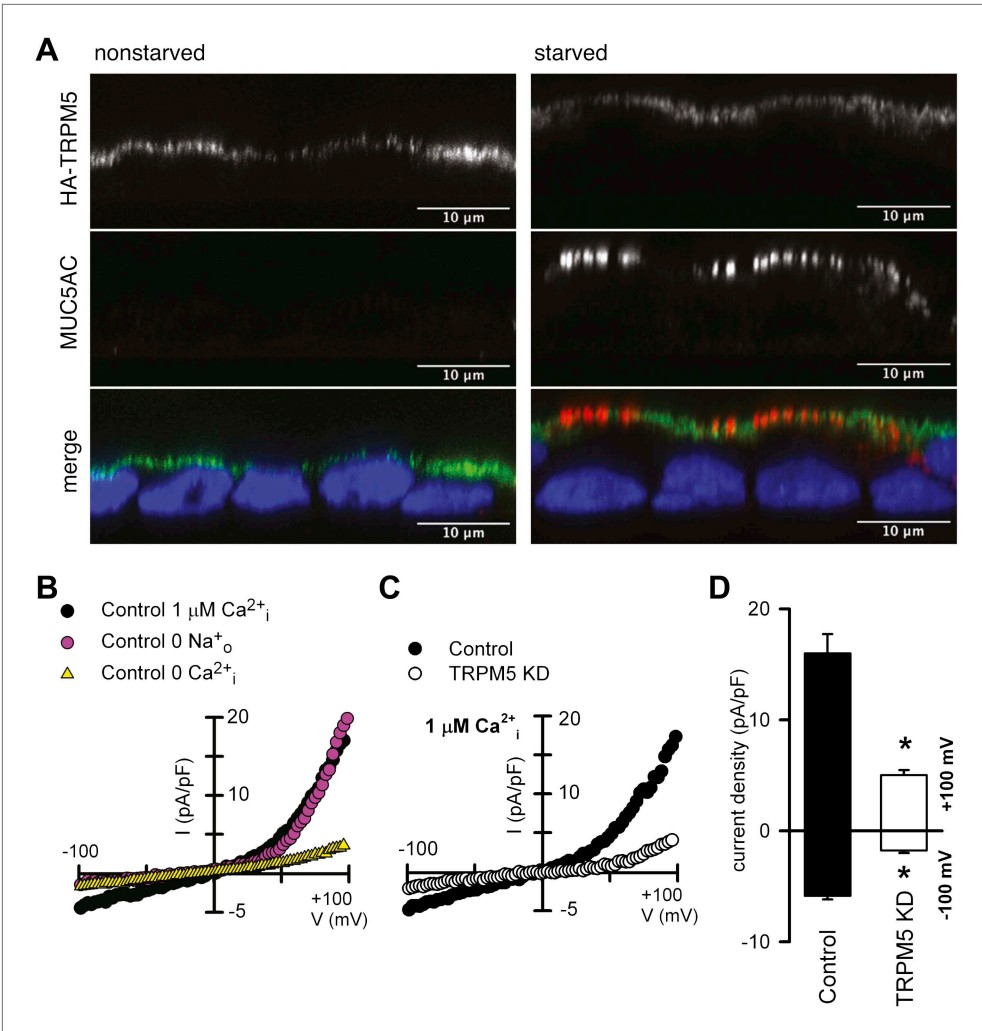

**Figure 6**. TRPM5 localization and activity. (**A**) Nonstarved and starved N2 cells stably transfected with HA-TRPM5 were fixed and analyzed by immunofluorescence microscopy with an anti-HA antibody (green) and an anti-MUC5AC antibody (red). The nuclear DNA was stained with DAPI (blue) to localize the position of the nucleus. (**B**) Ramp current-voltage relations of cationic currents recorded from a starved N2 cell dialyzed with internal solutions containing 1 µM $Ca^{2+}$ and bathed in $Na^+$-containing or $Na^+$-free solutions. A ramp current obtained in a cell dialyzed with internal solutions containing 0 $Ca^{2+}$ and bathed in $Na^+$-containing is also shown. (**C**) Representative ramp current-voltage relations of cationic currents recorded from a control and TRPM5-depleted N2 cells dialyzed with internal solutions containing 1 µM $Ca^{2+}$ and bathed in $Na^+$-containing solutions. (**D**) Mean TRPM5-like current density recorded at ±100 mV from control (n = 6) and TRPM5 KD cells (n = 8). * p<0.05.

secretion by differentiated N2 cells in the presence of extracellular $Ca^{2+}$ (**Figure 7B**). To further investigate the source of $Ca^{2+}$ in this process we measured MUC5AC secretion in cells treated with thapsigargin, which promotes the release of $Ca^{2+}$ from the ER (**Cantero-Recasens et al., 2010**) or in response to the $Ca^{2+}$ ionophore ionomycin, which mainly facilitates $Ca^{2+}$ entry into the cytoplasm from the extracellular medium (**Figure 7—figure supplement 1**). MUC5AC secretion by ionomycin was significantly higher than treatment with thapsigargin, however, these reagents were less effective, compared with ATP, in eliciting MUC5AC secretion. Together, these results indicate that the contribution of $Ca^{2+}$ release from internal stores is not as significant as $Ca^{2+}$ entry from the extracellular medium for MUC5AC secretion by N2 cells.

## TRPM5 modulates ATP-mediated $Ca^{2+}$ entry in goblet cells

Consistent with the secretion data, measurements of intracellular $Ca^{2+}$ levels in starved N2 cells loaded with the calcium dye Fura-2 showed that cells treated with ATP elicited an increase in intracellular

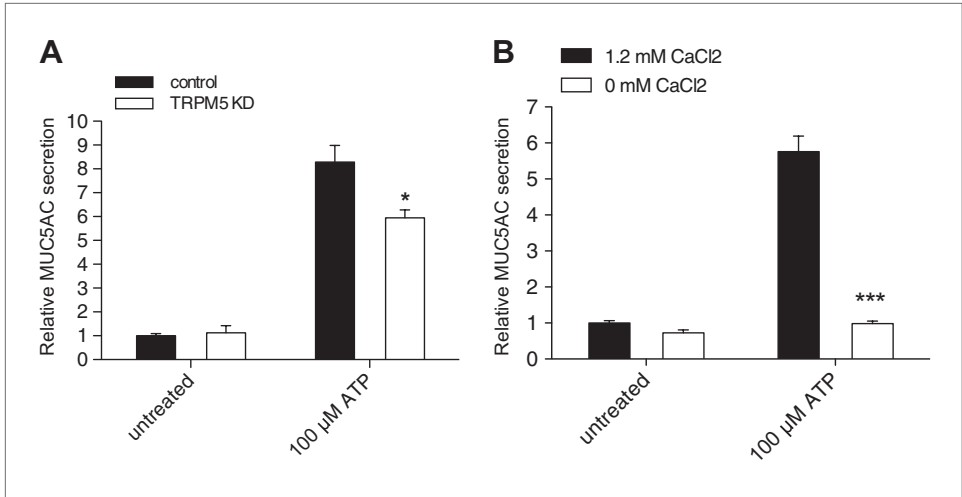

**Figure 7**. TRPM5 modulates ATP-induced MUC5AC secretion. (**A**) Control and TRPM5 stable knockdown cells were starved and incubated for 30 min at 37°C with 100 μM ATP. Secreted MUC5AC was collected and processed for dot blot analysis with an anti-MUC5AC antibody. The dot blots were quantified and normalized to intracellular actin levels. The y-axis represents relative values with respect to the values of untreated control cells. Average values ± SEM are plotted as bar graphs (N = 6). Datasets were considered as statistically significant when $p < 0.01$ (*). (**B**) Starved N2 cells were incubated for 30 min at 37°C with 100 μM ATP in the presence (1.2 mM $CaCl_2$) or absence (0 mM $CaCl_2$) of extracellular $Ca^{2+}$. Secreted MUC5AC was collected and analyzed by dot blot with an anti-MUC5AC antibody. The dot blots were quantified and normalized to intracellular actin levels. The y-axis represents relative values with respect to the values of untreated N2 cells in the presence of 1.2 mM $CaCl_2$. Average values ± SEM are plotted as bar graphs (N = 3). Datasets were considered as statistically significant when $p < 0.001$ (***).

The following figure supplements are available for figure 7:

**Figure supplement 1**. $Ca^{2+}$ dependent MUC5AC secretion.

$Ca^{2+}$ concentration, which was markedly decreased when extracellular $Ca^{2+}$ was removed (*Figure 8A*). Mean increases in the peak $Ca^{2+}$ signal are shown in the right panel. These data suggested a close link between secretagogue-induced $Ca^{2+}$ entry and MUC5AC secretion.

We then evaluated whether TRPM5 was required for the ATP-induced entry of extracellular $Ca^{2+}$ and the subsequent regulation of MUC5AC secretion. Control starved N2 cells and N2 cells stably depleted of TRPM5 were treated with 100 μM ATP and intracellular $Ca^{2+}$ levels were recorded. After ATP treatment, cells depleted of TRPM5 showed reduced increase in intracellular $[Ca^{2+}]$ compared with control cells (*Figure 8B*). To test if the reduction in intracellular $[Ca^{2+}]$ upon purinergic receptor activation with ATP reflected a defect in $Ca^{2+}$ influx from the extracellular medium, we measured the elevation in intracellular $Ca^{2+}$ level by ATP treatment in N2 cells and TRPM5-depleted cells in the absence of extracellular $Ca^{2+}$ (*Figure 8C*). In the absence of extracellular $Ca^{2+}$ there was no difference between control and TRPM5 depleted cells in ATP-induced increase of intracellular $Ca^{2+}$ levels, suggesting that TRPM5 participation in ATP-mediated MUC5AC secretion is related to the regulation of the secretagogue-induced $Ca^{2+}$ entry.

TRPM5 might be involved in modulating $Ca^{2+}$ influx by changing the cell membrane potential following the entry of monovalent cations. Positive modulation of $Ca^{2+}$ entry by TRPM5-mediated membrane depolarization has been linked to the activation of voltage-gated $Ca^{2+}$ channels (*Colsoul et al., 2010*; *Shah et al., 2012*). However, we detected neither voltage-gated whole-cell $Ca^{2+}$ currents (*Figure 9—figure supplement 1A*) nor depolarization-induced $Ca^{2+}$ signals (*Figure 9—figure supplement 1B*) in starved N2 cells. Accordingly, inhibitors of voltage-gated $Ca^{2+}$ channels did not modify ATP-mediated $Ca^{2+}$ signals (*Figure 9—figure supplement 1C*). Therefore, we hypothesized that TRPM5-mediated $Na^+$ entry was coupled to the functioning of a $Na^+/Ca^{2+}$ exchanger (NCX) in reverse mode, thereby promoting further $Ca^{2+}$ entry.

We investigated the participation of NCX in ATP-mediated MUC5AC secretion and $Ca^{2+}$ signaling using KB-R9743, an NCX inhibitor that preferentially blocks the reverse $Ca^{2+}$ influx mode of the

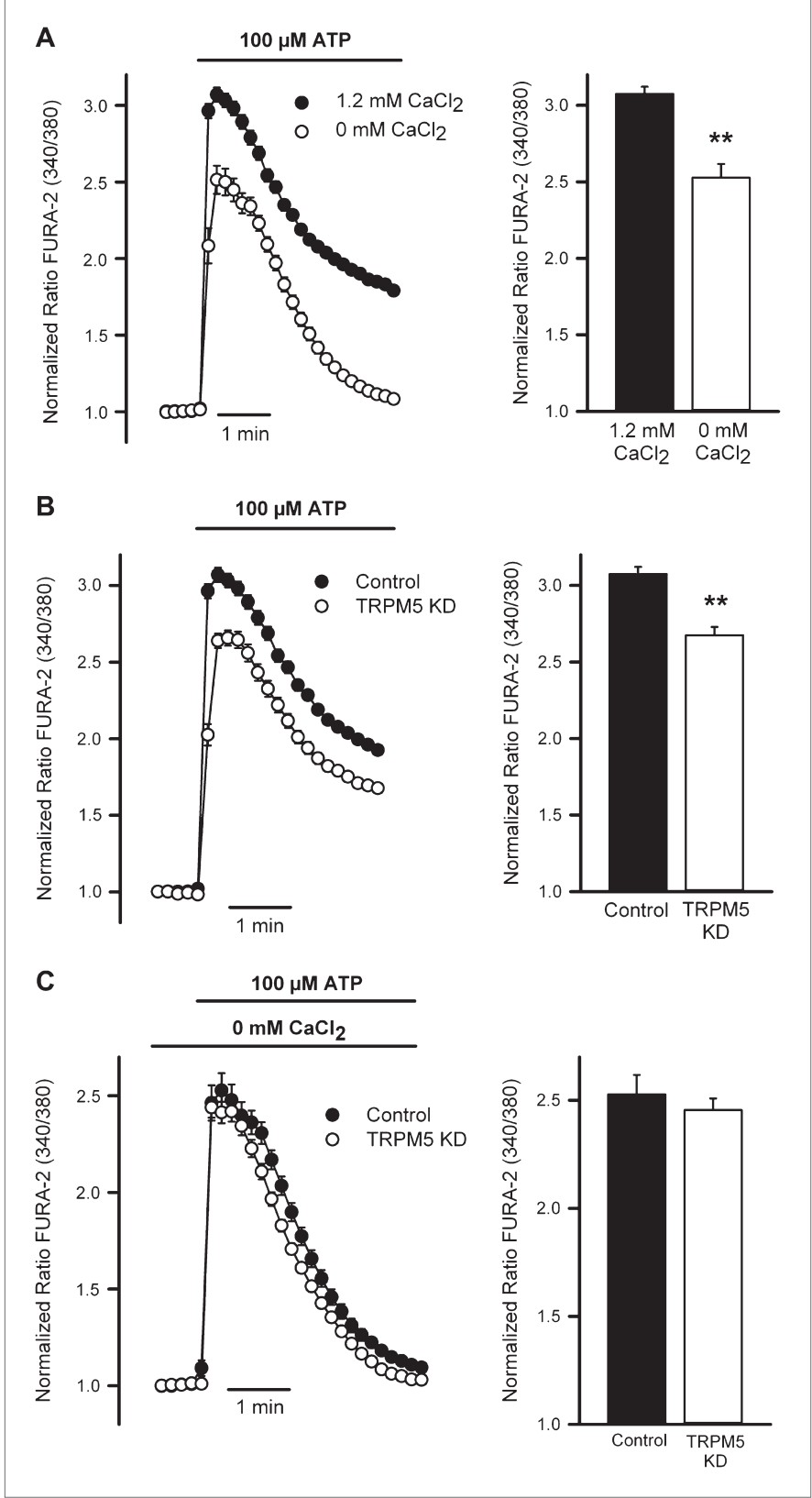

**Figure 8**. TRPM5 modulates ATP-induced Ca²⁺ entry. (**A**) Time course of mean Ca²⁺ responses (Fura-2 ratio) obtained in starved N2 cells treated with 100 μM ATP in the presence (n = 138) or absence of 1.2 mM Ca²⁺ (n = 118) in the extracellular solution. Right panel, average peak [Ca²⁺] increases obtained from traces shown in the right

*Figure 8. Continued on next page*

*Figure 8. Continued*

panel. *p<0.01. (**B**) Time course of mean $Ca^{2+}$ responses (Fura-2 ratio) obtained in starved control (n = 179) and TRPM5 KD N2 cells (n = 163) treated with 100 μM ATP. Right panel, average peak $[Ca^{2+}]$ increases obtained from traces shown in the right panel. *p<0.01. (**C**) Time course of mean $Ca^{2+}$ responses (Fura-2 ratio) obtained in starved control (n = 118) and TRPM5 KD N2 cells (n = 89) treated with 100 μM ATP and bathed in $Ca^{2+}$-free solutions. Right panel, average peak $[Ca^{2+}]$ increases obtained from traces shown in the right panel. *p<0.01.

transporter (*Iwamoto et al., 1996*). Control starved N2 cells and N2 cells stably depleted of TRPM5 were pretreated with 50 μM KB-R9743 for 15 min and then incubated with 100 μM ATP. ATP induced MUC5AC secretion was greatly reduced in the presence of the NCX inhibitor (*Figure 9A*), which suggests that TRPM5- and $Ca^{2+}$-dependent MUC5AC secretion involves the activity of an NCX. This hypothesis was further examined by measuring ATP-induced $Ca^{2+}$ signals in the presence of the NCX inhibitor. ATP-induced $Ca^{2+}$ signals were reduced by ~ 50% in cells treated with the NCX inhibitor (*Figure 9B*). Similar to the results obtained in the absence of extracellular $Ca^{2+}$ (*Figure 8D*), in the presence of the NCX inhibitor there was no difference in $Ca^{2+}$ signals between control and TRPM5-depleted N2 cells (*Figure 9B*). These results suggest that N2 cells exhibit an ATP-induced $Ca^{2+}$ entry mechanism that is consistent with the operation of an NCX in reverse mode and this control mechanism is lost in N2 cells depleted of TRPM5.

## Discussion

There are 17 different kinds of mucin genes and their products are either secreted or transported and inserted into the plasma membrane. The secreted gel-forming mucins MUC2, MUC5AC, MUC5B and MUC6 are produced by goblet cells, which are present in the epithelia and submucosal glands of the respiratory and gastrointestinal tract (*Thornton et al., 2008*; *McGuckin et al., 2011*). Surprisingly, human pathologies such as colon cancer and ulcerative colitis produce MUC5AC de novo, which is then secreted (*Bartman et al., 1999*; *Kocer et al., 2002*; *Forgue-Lafitte et al., 2007*; *Bu et al., 2010*). In general, mucins are produced as a result of cell differentiation and the newly synthesized mucins, like all other secretory proteins, are transported from the ER to the Golgi membranes. In the Golgi complex, the secreted forms of mucins are sorted and packed into granules; the granules mature, fuse with the plasma membrane, predominantly by the influx of $Ca^{2+}$ into the cells, and release their content. In cells of the gastro-intestinal lining (*Bou-Hanna et al., 1994*; *Barcelo et al., 2001*; *Bertrand et al., 2004*) and eye conjunctiva (*Li et al., 2012*) influx of extracellular $Ca^{2+}$ participates in the release of mucins from the secretory granules. $Ca^{2+}$-dependent events are also essential for the release of mucins from the respiratory tract, however, the source of $Ca^{2+}$ is unclear. The general view is that mucin secretion in the airways is dependent on $Ca^{2+}$ release from intracellular stores and independent of extracellular $Ca^{2+}$ (*Kemp et al., 2004*; *Davis and Dickey, 2008*). However, extracellular $Ca^{2+}$ is required for mucin secretion from cholinergic stimulated swine airway submucosal glands (*Lu et al., 2011*) as well as by cold and menthol stimulated human bronchial epithelial cells (*Li et al., 2011*). The involvement of extracellular $Ca^{2+}$ in mucin secretion is therefore likely to be cell type, signal, and mucin specific. The synthesis and secretion of mucins is controlled by a large number of distinct stimuli, which poses additional problems for the identification of proteins involved in mucin homeostasis (*Forstner et al., 1994*; *Stanley and Phillips, 1994*; *Epple et al., 1997*; *Slomiany and Slomiany, 2005*).

Overproduction and hyper secretion of gel-forming mucins is linked to COPD, asthma and cystic fibrosis (*Rose and Voynow, 2006*) and to the protection of the gut lining against infection and growth of a number of parasites including *H. pylori*. Inhibition of synthesis and secretion of mucins is linked to inflammatory bowel diseases such as ulcerative colitis and Crohn's disease (*Corfield et al., 2001*). The importance of understanding mucin synthesis and secretion is therefore more than just a scholarly exercise.

### Assay for measuring mucin secretion

The size and rheological properties of gel-forming mucins has hindered the development of a quantitative assay to monitor their secretion. Our antibody-based detection of secreted MUC5AC is relatively easy, quantitative, and highly accurate. It involves starvation-induced synthesis of MUC5AC, which is then released by treating the cells with PMA. It has recently been shown that secretion of total polymeric mucins from goblet-cell metaplastic human bronchial epithelial cultures is inhibited by BFA treatment

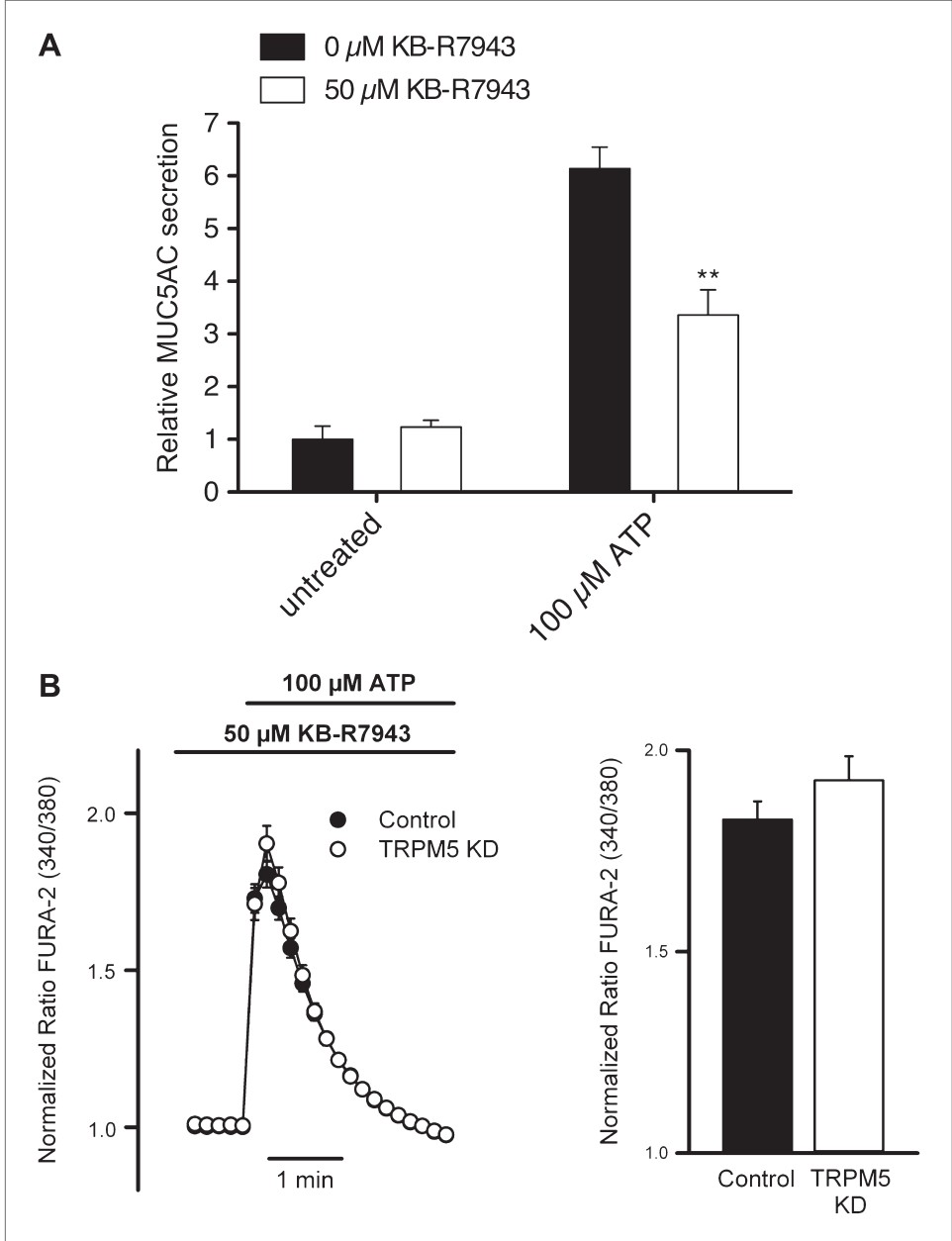

**Figure 9**. Effect of inhibiting the NCX on MUC5AC secretion and $Ca^{2+}$ entry. (**A**) Starved N2 cells were preincubated for 15 min with or without KB-R7943 (50 µM) followed by incubation with 100 µM ATP in the presence or absence of KB-R7943. Secreted MUC5AC was analyzed by dot blot with an anti-MUC5AC antibody. The dot blot was quantified and normalized to intracellular tubulin amount. The y-axis represents relative values with respect to values of untreated control cells. Average values ± SEM are plotted as bar graphs (N = 6). Datasets were considered as statistically significant when p<0.01 (**). (**B**) Time course of mean $Ca^{2+}$ responses (Fura-2 ratio) obtained in starved control (n = 84) and TRPM5 KD N2 cells (n = 83) treated with 100 µM ATP in the presence of 50 µM KB-R7943. Right panel, average peak [$Ca^{2+}$] increases obtained from traces shown in the right panel.

The following figure supplements are available for figure 9:

**Figure supplement 1**. Voltage-gated $Ca^{2+}$ channels are not expressed or functional in N2 cells.

(*Okada et al., 2010*). This likely represents secretion of newly synthesized mucin that is secreted at some basal rate. PMA mediated MUC5AC secretion reported here is unaffected by BFA treatment (*Figure 2D,E*). Our assay, therefore, measures release of MUC5AC from the post Golgi secretory storage granules.

## PIMS

Based on our experimental data from a pool of 7343 gene products tested, we selected 16 proteins because their knockdown significantly affected MUC5AC secretion from the goblet cell line. These proteins (PIMS) are expressed in the goblet cells and not required for general protein secretion. PIMS include ion channels and regulatory molecules (SUR1, GRIK4 and TRPM5); GPCR's (CCR9, GRP62 and CCBP2), transcription regulators (SREBF1, ATF6 and NFKB1), $Ca^{2+}$ binding protein (KCNIP3), GTPase activator for Rap1 that controls actin dynamics (SIPA1), actin binding protein (PLEK2), scaffold for the MAPK (TAB1), MAPK15, and a protein involved in melanosome biogenesis (SILV).

Actin dynamics are important for MUC5AC secretion and, as shown here, stablization of actin filaments but not their depolymerization inhibited MUC5AC secretion. The identification of SIPA1 and PLEK2 could help reveal the components involved in regulating Rap1, which is known to regulate actin filament dynamics in the events leading to the docking/fusion of the MUC5AC-containing secretory granules. SILV is required for the early stages of melanosome biogenesis, and goblet cells express SILV but are not known to make melanosomes. It is reasonable to propose that SILV performs an analogous function in the maturation of MUC5AC granules in the goblet cells. TAB1 and MAPK15 are likely involved in stress response-mediated synthesis and secretion of MUC5AC. The cell-surface ion channels and the GPCRs are likely involved in signaling events that lead to the secretion of MUC5AC. Future analysis of these proteins will help reveal their significance in MUC5AC homeostasis.

## TRPM5 and its role in regulated MUC5AC secretion

TRPM5 is a $Ca^{2+}$-activated monovalent cation selective channel that responds to warm temperature and a key component of the bitter, sweet and umami taste-receptor signaling cascade (*Perez et al., 2002*; *Zhang et al., 2003*). Bitter, sweet and umami tastants are detected by GPCRs that signal through gustducin and PLCβ2 in order to produce DAG and $IP_3$. $IP_3$ activates the release of $Ca^{2+}$ from the ER, which then activates TRPM5 (*Hofmann et al., 2003*; *Liu and Liman, 2003*). The activated TRPM5 affects membrane potential to control events leading to the sensation of bitterness. Interestingly, bitter taste receptors (GPCRs) are expressed in airway smooth muscle and GPCRs coupled to $G_{\alpha q}$ or $G_{\alpha s}$ control muscle contraction and relaxation, respectively (*Deshpande et al., 2010*). A large number of GPCRs regulate taste signaling, but it is not clear which family members are expressed in the goblet cells and linked to the activation of TRPM5. We have found several GPCRs and ion channels that are expressed in the goblet cells and required for MUC5AC secretion. It would be important to test whether any of these GPCRs are directly linked to TRPM5 activation in the goblet cells and therefore in the events leading MUC5AC secretion.

The activity of TRPM5, and its close relative TRPM4, has been shown to couple intracellular $Ca^{2+}$ to changes in membrane potential and either an activation of voltage-gated $Ca^{2+}$ channels favouring further $Ca^{2+}$ increase (*Earley et al., 2004*; *Colsoul et al., 2010*; *Uchida and Tominaga, 2011*; *Shah et al., 2012*) or a reduction in the electrochemical gradient for store-operated $Ca^{2+}$ entry (*Vennekens et al., 2007*). Our results provide evidence for a novel link between TRPM5 activity and the control of intracellular $Ca^{2+}$ signaling via the functional coupling of TRPM5 and NCX. NCX at the plasma membrane is one of the major means to extrude $Ca^{2+}$ (in exchange for $Na^+$) from the cells. However, this transporter can also function in the reverse mode to allow $Ca^{2+}$ entry when intracellular $Na^+$ concentration increases. Significant $Na^+$ entry is expected to occur during activation of TRPM5 following $IP_3$-mediated $Ca^{2+}$ release. Sodium entering through the TRPM5 would accumulate intracellularly, thus promoting NCX to act in the reverse mode and permit $Ca^{2+}$ entry in exchange for the exiting $Na^+$. Such a coupling between endogenous NCX and heterologously expressed $Na^+$ and $Ca^{2+}$ permeable TRPC3 channel has been reported (*Rosker et al., 2004*). However, our observation describes for the first time the functional coupling between endogenous TRPM5 channel and NCX, and its physiological relevance in the context of mucin secretion.

In conclusion, we have devised a quantitative assay to measure mucin secretion in human goblet cell lines. Our findings have revealed new components that control MUC5AC secretion. We have described the mechanism linking the activity of TRPM5 with the modulation of $Ca^{2+}$ signals and

MUC5AC secretion from the differentiated colonic goblet cell line. It is important now to test the expression of PIMS in cells of the respiratory and gastric lining; their involvement in secretion of MUC5AC and other secreted mucins such as MUC2 to understand the mechanisms of mucin homeostasis.

## Materials and methods

### Reagents and antibodies

All chemicals were obtained from Sigma-Aldrich (St. Louis, MO) except BAPTA-AM (Biomol International, Farmingdale, NY), Latrunculin A (Calbiochem, Billerica, MA), Jasplakinolide (Life Technologies, Carlsbad, CA) and KB-R7943 (Tocris Bioscience, Bristol, UK), anti-MUC5AC antibody clone 45M1 (Neomarkers, Waltham, MA), anti-GRASP65 C-20 antibody from Santa Cruz Biotechnology (Dallas, TX), anti-actin antibody clone AC-15 from Sigma-Aldrich anti-β tubulin antibody from Sigma-Aldrich and anti-HA antibody clone 3f10 from Roche (Basel, Switzerland). Secondary antibodies for immunofluorescence microscopy and dot blotting were from Life Technologies.

### Generation of plasmids

The shRNA targeting TRPM5 was constructed as follows: target sequence 5'- GTACTTCGCCTTCC TCTTC - 3', loop 5' - TTCAAGAGA - 3', antisense 5' - GAAGAGGAAGGCGAAGTAC - 3', terminator 5' - TTTTTTC - 3'. The sense and antisense shRNA oligos were annealed and cloned via HpaI and XhoI into the lentilox 3.1 vector containing GFP and sequence verified. HA-pcDNA3.1 plasmid was constructed as follows: primers encoding the HA-tag were designed (forward primer 5' - GGCCGCACC ATGTACCCTTACGACGTTCCTGATTACGCATCCCTTGAATTCCCCGGGG - 3' and reverse primer 5' - GAT CCCCCGGGGAATTCAAGGGATGCGTAATCAGGAACGTCGTAAGGGTACATGGTGC - 3'), annealed, cloned via NotI and BamHI into pcDNA3.1 and sequence verified. TRPM5 cDNA was obtained from Imagenes (UK) (accession number BC093787) and amplified using the following primers: forward primer 5' - CGCATAGAATTCCAGGATGTCCAAGGCCCCCG - 3' and reverse primer 5' - TAAGGTACCTGTTCA GGTGTCCGAGGGAGGCTGGCC. The amplified PCR product was cloned into HA-pcDNA3.1 plasmid via EcoRI and KpnI and sequence verified. HA-TRPM5 was cloned via NotI and BclI into a lentiviral vector. All primers were obtained from Sigma-Aldrich.

### Differentiation of N2 cells

N2 cells were seeded in complete growth medium (DMEM complemented with 10% FBS). The next day (d0) medium was exchanged for PFHMII protein free medium (Invitrogen, Carlsbad, CA) and cells were grown for 3 days. At day 3 (d3) the medium was replaced with fresh PFHMII medium and the cells were grown for 3 additional days (total time 6 days). At day 6 (d6) cells were seeded in complete growth medium for different experiments following replacement of complete growth medium for PFHMII medium at day 7 (d7) and experimental procedure at day 9 (d9).

### Mucin secretion assay

N2 cells were differentiated for 6 days and then aliquoted into the wells of a 96-well plate. After 1 day of growth (d7), the medium was replaced by fresh PFHMII medium and the cells were grown for 2 more days (total time 9 days). On d9 cells were treated with 2 μM PMA for 2 hr at 37°C. The secreted MUC5AC was fixed by adding paraformaldehyde (PFA) to the cells at a final concentration of 4% for 30 min at RT. Cells were subsequently washed with PBS and incubated with the anti-MUC5AC antibody (clone 45M1) diluted 1:100 in 4% BSA/PBS for 1 hr at RT. The cells were then washed with PBS and incubated with a donkey anti-mouse HRP conjugated antibody (Santa Cruz Biotechnology) diluted 1:10,000 in 4% BSA/PBS for 1 hr at RT. After washing the cells in PBS, 100 μl ECL solution was added to the cells and luminescence was measured with a Tecan plate reader.

### Dot blot analysis

N2 cells were differentiated for 6 days and then seeded into six-well plates. The cells were processed until d9 as described for the mucin secretion assay. On d9 cells were treated with 100 μM ATP for 30 min at 37°C. Supernatant was collected and centrifuged for 5 min at 800×$g$ at 4°C. Cells were washed 2× in PBS and lysed in 1% Triton X-100/1 mM DTT/PBS for 1 hr at 4°C and centrifuged at 1000×$g$ for 10 min. The supernatants and cell lysates were spotted on nitrocellulose membranes and membranes were incubated in blocking solution (4% BSA/0.1% Tween/PBS) for 1 hr at room temperature. The blocking solution was removed and the membranes were incubated with the anti-MUC5AC antibody

diluted 1:1000 or the anti-actin antibody at a dilution of 1:1000 in blocking solution. Membranes were washed in 0.1% Tween/PBS and secondary antibodies conjugated to HRP were incubated in blocking solution at a dilution of 1:10,000 for 1 hr at room temperature. For the detection of β-tubulin, cell lysates were separated on SDS-PAGE, transferred to nitrocellulose membranes and processed as described for the dot blot analysis using the anti-β-tubulin antibody at a dilution of 1:10,000. Membranes were washed, incubated with ECL substrate and imaged with a Fujifilm LAS-3000 camera. Membranes were analyzed and quantitated in ImageJ (version 1.44o; National Institutes of Health).

## Screen procedure and data analysis

N2 cells were differentiated for 6 days. On d6, $4 \times 10^4$ cells were seeded into the wells of a 96-well plate and transfected in triplicates on three sets of plates with 150 nM siRNA (provided by the high throughput screening facility at the Center for Genomic Regulation) and Dharmafect 4 (Dharmacon, Lafayette, CO) according to manufacturer's instructions. The cells grown on the plates were handled until d9 as described above. On d9, cells were treated with 2 µM PMA for 2 hr at 37°C and processed for MUC5AC secretion as described in the Mucin secretion assay. The Mucin secretion assay was automated and performed on the Caliper LS staccato workstation. Each plate was normalized by the B-score method (*Brideau et al., 2003*) and positive hits were selected above B-score 1.5 and below B-Score −1.5. The hits were classified using the ranking product method (*Breitling et al., 2004*) using the triplicates. The data was analyzed and automated by a script written with the statistical toolbox from Matlab (Mathwork). The validation screen was performed exactly as described for the screen procedure. The ontarget PLUS siRNAs were obtained from Dharmacon (Lafayette, CO). All the plates were normalized platewise by:

$$z\text{-score} = \left(xi - average(xn)\right)/SD(xn),$$

xn = total population and xi = sample. Positive hits were selected 2 SD above and below mock treated samples.

## Immunofluorescence analysis

Undifferentiated and differentiated N2 cells were grown on coverslips. For the visualization of intracellular MUC5AC cells were fixed with 4% PFA/PBS for 30 min at RT. Cells were washed with PBS and permeabilized for 20 min with 0.2% Triton X-100 in 4% BSA/PBS. The anti-MUC5AC antibody was added to the cells at 1:1000 in 4% BSA/PBS for 1 hr. Cells were washed in PBS and incubated with a donkey anti-mouse Alexa 488 coupled antibody (Invitrogen), diluted at 1:1000 in 4% BSA/PBS, and DAPI. Cells were washed in PBS and mounted in FluorSave Reagent (Calbiochem, Billerica, MA). For the detection of secreted MUC5AC, differentiated N2 cells were treated with 2 µM PMA for 2 hr at 37°C. The secreted MUC5AC was fixed on the cells by adding PFA to the cells at a final concentration of 4% for 30 min at RT. The cells were then processed for immunofluorescence analysis (as described before) without the permeabilization step with Triton X-100. For the removal of secreted MUC5AC, cells were incubated for 2 hr with 2 µM PMA at 37°C. The cells were then placed on ice and washed 2× with ice cold PBS. Subsequently, cells were incubated in 1 mM DTT/0.05% Trypsin-EDTA 1× (Invitrogen)/PBS for 10 min at 4°C, following four washes in ice-cold PBS and two washes in 4% BSA/PBS. The cells were then fixed in 4% PFA/PBS for 30 min at room temperature, permeabilized with 0.2% Triton X-100 in 4% BSA/PBS and processed for immunofluorescence as described before. Cells were imaged with a confocal microscope (SP5; Leica) using the 63× Plan Apo NA 1.4 objective. For detection, the following laser lines were applied: DAPI, 405 nm; and Alexa Fluor 488, 488 nm; Alexa Fluor 568, 561 nm. Pictures were acquired using the Leica software and converted to TIFF files in ImageJ (version 1.44o; National Institutes of Health).

## Pulse chase experiment

Differentiated N2 cells grown on six-well plates were starved in methionine- and cystine-free DMEM (Invitrogen, Carlsbad, CA) for 20 min at 37°C. Cells were labeled with 100 µCi 35 S-methionine for 15 min and chased for 3 hr at 37°C in medium supplemented with 10 mM L-methionine. Brefeldin A (BFA) Sigma-Aldrich was added at a concentration of 2 µg/ml during starvation, pulse and chase. The supernatant was collected and centrifuged for 5 min at 800×*g* at 4°C. Cells were washed with PBS and lysed in 1% Triton X-100/PBS for 1 hr at 4°C, following centrifugation for 30 min at 4°C at 16,000×*g*. Lysates were measured for [35]S-methionine incorporation with a beta-counter. Supernatants

were normalized to incorporated $^{35}$S-methionine and precipitated by TCA. Samples were separated by SDS-PAGE and analyzed by autoradiography.

## Measuring expression profile

Unstarved- and 5-day starved N2 cells were lysed and total RNA was extracted with the RNeasy extraction kit (Qiagen, Netherlands). Total RNA was treated with Dnase I (New England Labs, Ipswich, MA) for 1 hr at 37°C and purified by phenol extraction. cDNA was synthesized with Superscript III (Invitrogen). Primers for each gene (sequence shown below, *Table 3*) were designed using Primer 3 v 0.4.0 (*Rozen and Skaletsky, 2000*), limiting the target size to 300 bp and the annealing temperature to 60°C. To determine expression levels of MUC5AC and TRPM5, quantitative real-time PCR was performed with Light Cycler 480 SYBR Green I Master (Roche, Switzerland) according to manufacturer's instructions. Expression of PIMS in unstarved and starved cells was determined by quantifying the PCR band intensities with ImageJ software.

## Generation of stable shRNA knockdown cell lines

Lentivirus was produced by co-tranfecting HEK293 cells with the plasmid, VSV.G and delta 8.9 by calcium phosphate. At 48 hr posttransfection the secreted lentivirus was collected, filtered and directly added to N2 cells. Stably infected cells were either selected by puromycine resistance or sorted for GFP positive signal by FACS.

## Electrophysiology recordings

The whole-cell configuration of the patch-clamp technique was employed as previously describe to test for the functional expression of TRP channel activity (*Fernandes et al., 2008*) and voltage-gated calcium currents (*Serra et al., 2010*). Pipettes with a resistance of 2–3 MΩ were used. To record voltage-gated calcium currents we employed an external solution containing (in mM): 140 tetraethylammonium-Cl, 3 CsCl, 5 CaCl$_2$, 1.2 MgCl$_2$, 10 HEPES and 10 glucose (pH 7.4 adjusted with Tris); and an intracellular solution: 140 CsCl, 1 EGTA, 4 Na$_2$ATP, 0,1 Na$_3$GTP and 10 HEPES (pH 7.2–7.3 adjusted with Tris; and 295–300 mosmoles/l). To record TRPM5 currents we used an external solution containing: 140 NaCl, 2.5 KCl, 0.5 MgCl$_2$, 5 Glucose, 10 Hepes, pH was adjusted to 7.4 with Tris, and 300–305 mosmoles/l; and

**Table 3.** Primer sequences used for detecting mRNA for the respective PIMS in N2 cells

| Gene | Forward primer 5′–3′ | Reverse primer 5′–3′ |
|------|----------------------|----------------------|
| TRPM5 | GTGGCCATCTTCCTGTTCAT | CTTCATCATGCGCTCTACCA |
| CCR9 | GCCAGCCTTGGCCCTGTTGT | TCCAGCAAGGCCTGCGCTTC |
| PLEK2 | AGAACAGGCCAGTGGGTGGGT | GCTCGCTCAGCCTTGCTGCT |
| TAB1 | TCAATCATCGCAGCAATCTC | GGCTACACGGACATTGACCT |
| KCNIP3 | CCACCACCTATGCACACTTCC | GTCGTAGAGATTAAAGGCCCAC |
| SILV | GGGCTACAAAAGTACCCAGAAAC | CCTTGAGGGACACTTGACCAC |
| SIPA1 | CTCCTTTCTGCCACGTACCTT | TTTTGGAGTTCCCTTAGGGTCT |
| HPRT1 | TGACACTGGCAAAACAATGCA | GGTCCTTTTCACCAGCAAGCT |
| MUC5AC | CAACCCCTCCTACTGCTACG | CTGGTGCTGAAGAGGGTCAT |
| GPR62 | GGTGGTTTCCGTGGGGGCTC | TGGGCCCAGACCGCAGGATT |
| PAG1 | TGGACGGCAGCCATGCATCC | ACTGTTGGTGTGGGCAGCGG |
| ATF6 | AGGTGGGTAGCGGTTGGGAGG | GCGGCACCTTACAGGCACCC |
| SREBF1 | CCACGGCAGCCCCTGTAACG | GGGACTGAGACCTGCCGCCT |
| MAPK15 | TACAACAGGTCCCTCCCCGGC | CCCAGTGCCGAGTGGCAGAC |
| SUR1 | GCCTTCGCAGACCGCACTGT | CTGCACGGACGAAGGAGGCG |
| NFKB1 | CGCCACCCGGCTTCAGAATGG | GTATGGGCCATCTGCTGTTGGCA |
| CCBP2 | CGGCGGGCATGGGACCATTT | AAGGCCACCACCAAGGCTGC |
| GRIK4 | CGTGGCTCGTGATGGTCGCC | GCCTCTCAGGAGCGCGGTTG |
| GAPDH | TGCACCACCAACTGCTTAGC | GGCATGGACTGTGGTCATGAG |

internal solution: 140 Cs acetate, 0.85 $CaCl_2$, 2 $MgCl_2$, 10 HEPES, 2 ATP, 0.5 GTP, pH 7.2 adjusted with Tris. Free intracellular calcium concentration to record TRPM5 current was adjusted to either 1 µM or <50 nM (0 Ca solution) with EGTA as calculated with WEBMAXC (http://www.stanford.edu/~cpatton/ webmaxcS.htm).

Cells were plated in 35-mm plastic dishes and mounted on the stage of an Inverted Olympus IX70 microscope. Whole cell currents were recorded with an Axon200A amplifier or with a D-6100 Darmstadt amplifier, filtered at 1 kHz. Currents were acquired at 33 kHz. The pClamp8 software (Axon Instruments, Foster City, CA) was used for pulse generation, data acquisition and subsequent analysis. Cells were clamped at −80 mV and pulsed for 20 ms from −60 mV to +60 mV in 5 mV steps when recording voltage-gated $Ca^{2+}$ currents or clamped at 0 mV and applying ramps from −100 mV to +100 mV (400 ms) at 0.2 Hz to record TRPM5 currents.

## Measurement of intracellular [$Ca^{2+}$]

Cells were plated onto glass coverslips, loaded with 5 µM of Fura-2AM for 30 min at room temperature, washed out thoroughly and bathed in an isotonic solution containing (in mM): 140 NaCl, 2.5 KCl, 1.2 $CaCl_2$, 0.5 $MgCl_2$, 5 glucose, 10 HEPES (305 mosmol/l, pH 7.4 adjusted with Tris). $Ca^{2+}$-free solutions were obtained by replacing $CaCl_2$ with equal amount of $MgCl_2$ plus 0.5 mM EGTA. ATP was added to the bath solution as indicated in the figure legend. All experiments were carried out at room temperature as previously described (*Fernandes et al., 2008*). AquaCosmos software (Hamamatsu Photonics) was used for capturing the fluorescence ratio at 505 nm obtained post-excitation at 340 and 380 nm. Images were computed every 5 s.

## Acknowledgements

Vivek Malhotra is an Institució Catalana de Recerca i Estudis Avançats (ICREA) professor at the Center for Genomic Regulation in Barcelona. The lentiviral system was kindly provided by Prof Thomas Graf. The screen was carried out at the Biomolecular Screening & Protein Technologies Unit at Centre Regulacio Genomica (CRG), Barcelona. Cell sorting experiments were carried out by the joint CRG/ UPF FACS Unit at Parc de Recerca Biomèdica de Barcelona. Fluorescence microscopy was performed at the Advanced Light Microscopy Unit at the CRG, Barcelona. Thanks to Anja Leimpek for technical assistance during the screening. Members of the Malhotra laboratory are thanked for valuable discussions.

## Additional information

### Competing interests

VM: Reviewing editor, *eLife*. The other authors declare that no competing interests exist.

### Funding

| Funder | Grant reference number | Author |
| --- | --- | --- |
| Sandler Foundation for Asthma Research | | Vivek Malhotra |
| Institucio Catalana de Recerca i Estudis Avancats (ICREA) | | Vivek Malhotra |
| Plan Nacional | BFU2008-00414 | Vivek Malhotra |
| Consolider | CSD2009-00016 | Vivek Malhotra |
| Agencia de Gestio d' Ajuts Universitaris i de Recerca (AGAUR) Grups de Recerca Emergents | SGR2009-1488 | Vivek Malhotra |
| European Research Council | 268692 | Vivek Malhotra |
| ICREA Academia Award | | Miguel A Valverde |
| Spanish Ministry of Economy and Competitiveness | SAF2012-38140 | Miguel A Valverde |
| Spanish Ministry of Economy and Competitiveness | SAF2012-31089 | José M Fernández-Fernández |

| Funder | Grant reference number | Author |
|---|---|---|
| Fondo de Investigacion Sanitaria | Red HERACLES RD12/0042/0014 | Miguel A Valverde |
| FEDER Funds | | Miguel A Valverde |
| SGR09-1369 | | Miguel A Valverde |

The funders had no role in study design, data collection and interpretation, or the decision to submit the work for publication.

## Author contributions

SM, MAV, Conception and design, Acquisition of data, Analysis and interpretation of data, Drafting or revising the article; CN, Acquisition of data, Analysis and interpretation of data, Drafting or revising the article; GC-R, KK, J-FP, LC, Development of experimental procedures, Acquisition of data, Drafting or revising the article; JMF, RG, Development of experimental procedures, Acquisition of data, Analysis and interpretation of data, Drafting or revising the article; FAB, Development of experimental procedures, Conception and design, Acquisition of data, Drafting or revising the article; VM, Conception and design, Analysis and interpretation of data, Drafting or revising the article

## Additional files

### Supplementary files

• Supplementary file 1. List of positive hits from primary screen. Lists positive hits from the primary screen, their reason for exclusion, the B-score, the ranking product and the E-value.

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
