## [Decision Letter]

Thank you for sending your work entitled “TRPM5-mediated calcium uptake regulates mucin secretion from human colon goblet cells” for consideration at *eLife*. Your article has been favorably evaluated by a Senior editor and 3 reviewers, one of whom is a member of our Board of Reviewing Editors.

The Reviewing editor and the other reviewers discussed their comments before we reached this decision, and the Reviewing editor has assembled the following comments to help you prepare a revised submission.

Your work presents an investigation of the genetic and physiological requirements for regulated secretion of Mucin 5AC from a differentiated human goblet cell line. The mucin coat is a key protective element of respiratory and intestinal epithelia, and the signals and mechanisms that underlie its synthesis and maintenance are poorly understood. Hence, the work addresses a very important question. Genetic requirements for Mucin 5AC secretion were elucidated with a siRNA-based screen of PMA-induced N2 cells that detects secreted MUC5AC. The reviewers agreed that the screen and hit validation process was rigorously executed, so there is high confidence that the PIMS gene products are key players in Muc5AC secretion, at least in the context of the N2 cell line. The second half of the manuscript uses electrophysiological experiments to address the specific role of one of the hits, TRPM5, in MUC5AC secretion. The results make a convincing case that TRPM5 plays a key and previously unappreciated role in NCX-mediated calcium influx (reverse mode) required for MUC5AC secretion, at least in a colonic cell line. However, the reviewers had three substantive concerns:

1) The Introduction and Discussion focused on the pathophysiological importance of airway mucin secretion, but you used a different cell type (colonic epithelium) from the one of interest (airway epithelium; Rossi AH, Am J Physiol 2007, 292:92; Kemp PA, Am J Respir Cell Mol Biol 2004, 31:446). Importantly, MUC5AC is not normally produced by colonic epithelium, so the physiological relevance of the results is unclear. This problem was compounded by a number of misstatements. For example, it is stated that secreted mucins “coat the plasma membranes”, which is true in the gut but not in the airway. In addition, “mucus coats epithelial surfaces exposed to the external environment”: this is true (with the caveat above) of some surfaces such as the airways, gut and, genitourinary tract, but not the skin. It is also stated that MUC5AC and MUC5B are the major components of mucus, which is true of the airway but not the gut, as noted above. A small point is that Muc5ac should be used for the non-human protein and MUC5AC for the human, rather than Muc5AC for both.

2) There has been extensive work on airway epithelia showing that calcium entry from outside the cell is not important acutely in secretion (Bahra P, Brit J Pharmacol 2004, 143:91; Scott CE, Am J Physiol 1998, 275:C285; Rossi AH, J Physiol 2004. 559:555). Since the authors speculate that reduced calcium entry following TRPM5 knockdown is a major mechanism, what is the relevance of the findings to airway epithelium? Are mechanisms of calcium handling different between colonic and airway epithelium?

The reviewers strongly recommend that these two concerns should be remedied by extensive rewriting. The Introduction and Discussion should include a careful analysis of potential differences between colonic and airway epithelial cells, and their mechanisms of induction of mucin secretion.

3) The reviewers were concerned that the 2hr PMA stimulation could introduce artifacts. How was this time determined and is it the optimal time for induction? Is it in the linear response range? Since mucin is present in intracellular vesicles that are primed for release, a much shorter activation time would seem more physiological. A 2hr treatment would allow new protein translation and possibly transcription. Therefore, have the authors examined whether addition of cycloheximide, or even APC, during PMA treatment affects mucin release? These experiments might provide insight into the role of the hits identified in the screen (are they expressed constitutively for mucin release, or expressed upon physiological stimulation?).

---

## [Author Response]

We want to state at the outset that we have used a colonic cancer cell line because it is easy to culture, easy to transfect, and it expresses and secretes MUC5AC upon differentiation. Altogether, this is a useful approach to identify proteins required for the secretion of MUC5AC. This is similar, in principle, to the use of HeLa cells and VSV-G to study the components of the secretory pathway. And like the HeLa cells-VSV-G combination, HT29-18N2 cells-MUC5AC has revealed a number of new components in the PMA mediated, post-Golgi release of MUC5AC from the secretory granules. BFA does not affect the secretion of MUC5AC in our experimental conditions, which gives us the confidence to rule out the effects of PIMS on the synthesis, export, and sorting of MUC5AC at the TGN. The next obvious step is to test the involvement of PIMS in primary cultures of the cells that line the respiratory and digestive tract. We expect many of the PIMS will be required for the secretion of MUC5AC and other secreted mucins in other cell types, but there could be noticeable differences as well. Some of these proteins might have cell type specific isoforms or closely related proteins that function at the same step in other cell types. Identification of PIMS will help address many of these issues as we learn more about the system.

*1) The Introduction and Discussion focused on the pathophysiological importance of airway mucin secretion, but you used a different cell type (colonic epithelium) from the one of interest (airway epithelium; Rossi AH, Am J Physiol 2007, 292:92; Kemp PA, Am J Respir Cell Mol Biol 2004, 31:446). Importantly, MUC5AC is not normally produced by colonic epithelium, so the physiological relevance of the results is unclear. This problem was compounded by a number of misstatements. For example, it is stated that secreted mucins “coat the plasma membranes”, which is true in the gut but not in the airway. In addition, “mucus coats epithelial surfaces exposed to the external environment”: this is true (with the caveat above) of some surfaces such as the airways, gut and, genitourinary tract, but not the skin. It is also stated that MUC5AC and MUC5B are the major components of mucus, which is true of the airway but not the gut, as noted above. A small point is that Muc5ac should be used for the non-human protein and MUC5AC for the human, rather than Muc5AC for both*.

We have addressed this issue by rewriting the Abstract and Introduction. The Abstract now begins with the following statement: “Mucin 5AC (MUC5AC) is secreted by goblet cells of the respiratory tract and, surprisingly, also expressed de novo in mucus secreting cancer lines.”

The Introduction begins with the following statements: “Mucus is secreted by specialized cells that line the respiratory and digestive tract to protect against pathogens and other forms of cellular abuse. The secretion of mucus is therefore essential for the normal physiology of the wet mucosal epithelium (52). The secretory or gel-forming mucin Mucin 5AC (MUC5AC) is one of the major components of the mucus in the airways, and hyper- or hyposecretion of this component is a hallmark of a number of chronic obstructive pulmonary diseases (COPD) (48). MUC5AC is also expressed at low levels in the gastrointestinal tract and, surprisingly, expressed de novo, and upregulated in colonic mucus from cancer and ulcerative colitis patients (7; 35; 14; 26; 13).”

The last paragraph of the Introduction is rewritten to address the reviewers concern with respect to the use of colon cell lines to address the mechanism of MUC5AC secretion. The changes in the text follow:

“As stated above, human cancer cells and cells from patients with ulcerative colitis express and secrete MUC5AC. These cells and cell lines therefore provide a convenient means to address the mechanism MUC5AC secretion. We have established a quantitative assay to measure the secretion of MUC5AC from a human goblet cell line. The procedure was used to screen 7343 human gene products and we describe here the identification and involvement of transient receptor potential melastatin 5 (TRPM5) channel in MUC5AC secretion.”

We now include the following statements in the final paragraph of the Discussion: “It is important now to test the expression of PIMS in cells of the respiratory and gastric lining, their involvement in secretion of MUC5AC, and other secreted mucins such as MUC2, to understand the mechanisms of mucin homeostasis.”

*2) There has been extensive work on airway epithelia showing that calcium entry from outside the cell is not important acutely in secretion (Bahra P, Brit J Pharmacol 2004, 143:91; Scott CE, Am J Physiol 1998, 275:C285; Rossi AH, J Physiol 2004. 559:555). Since the authors speculate that reduced calcium entry following TRPM5 knockdown is a major mechanism, what is the relevance of the findings to airway epithelium? Are mechanisms of calcium handling different between colonic and airway epithelium*?

We now state the following in the Discussion: “In cells of the gastro-intestinal lining (9; 6; 8) and eye conjunctiva (36) influx of extracellular Ca^2+^ participates in the release of mucins from the secretory granules. Ca^2+^-dependent events are also essential for the release of mucins from the respiratory tract; however, the source of Ca^2+^ is unclear. The general view is that mucin secretion in the airways is dependent on Ca^2+^ release from intracellular stores and independent of extracellular Ca^2+^ (34; 18). However, extracellular Ca^2+^ is required for mucin secretion from cholinergic stimulated swine airway submucosal glands (40) as well as by cold and menthol stimulated human bronchial epithelial cells (37). The involvement of extracellular Ca^2+^ in mucin secretion is therefore likely to be cell type, signal, and mucin specific.”

Additionally, the references cited by the reviewers do not address the role of Ca^2+^ entry in mucin secretion. Scott 1998 and Rossi 2004, respectively, reported that mucin secretion required both PKC (Ca^2+^ independent) and Ca^2+^ dependent components in an airway epithelial cell line SPOC1 permeabilized with O-streptolysin. Bahra and colleagues (2004) did not measure mucin secretion: they analysed the calcium responses of P2Y2 receptors in human bronchial epithelial cells and reported the involvement of a Ca^2+^ influx component.

*The reviewers strongly recommend that these two concerns should be remedied by extensive rewriting. The Introduction and Discussion should include a careful analysis of potential differences between colonic and airway epithelial cells, and their mechanisms of induction of mucin secretion*.

The inclusion of the text, as above, makes it clear that the source of Ca^2+^ in mucin secretion is cell type specific and currently a controversial issue.

*3) The reviewers were concerned that the 2 hr PMA stimulation could introduce artifacts. How was this time determined and is it the optimal time for induction? Is it in the linear response range? Since mucin is present in intracellular vesicles that are primed for release, a much shorter activation time would seem more physiological. A 2 hr treatment would allow new protein translation and possibly transcription. Therefore, have the authors examined whether addition of cycloheximide, or even APC, during PMA treatment affects mucin release? These experiments might provide insight into the role of the hits identified in the screen (are they expressed constitutively for mucin release, or expressed upon physiological stimulation?)*.

We now show a time course for the PMA dependent secretion of MUC5AC. The time course for PMA induced MUC5AC secretion revealed a significant increase of MUC5AC secretion at 15 min and maximal MUC5AC secretion was observed after 2h incubation with 2 µM PMA (Figure 2—figure supplement 1).

It is also very important to note that PMA mediated MUC5AC release is not inhibited by BFA. The release of MUC5AC in our experimental conditions, therefore, is from a post-Golgi pool. This alleviates any potential effects on the synthesis of MUC5AC.